# The Two-Component System 09 Regulates Pneumococcal Carbohydrate Metabolism and Capsule Expression

**DOI:** 10.3390/microorganisms9030468

**Published:** 2021-02-24

**Authors:** Stephanie Hirschmann, Alejandro Gómez-Mejia, Ulrike Mäder, Julia Karsunke, Dominik Driesch, Manfred Rohde, Susanne Häussler, Gerhard Burchhardt, Sven Hammerschmidt

**Affiliations:** 1Department of Molecular Genetics and Infection Biology, Interfaculty Institute for Genetics and Functional Genomics, Center for Functional Genomics of Microbes, University of Greifswald, 17487 Greifswald, Germany; hirschmans@uni-greifswald.de (S.H.); alejandro.gomezmejia@usz.ch (A.G.-M.); julia.karsunke@gmx.de (J.K.); gerhard.burchhardt@uni-greifswald.de (G.B.); 2Department of Functional Genomics, Interfaculty Institute for Genetics and Functional Genomics, Center for Functional Genomics of Microbes, University Medicine Greifswald, 17475 Greifswald, Germany; ulrike.maeder@uni-greifswald.de; 3BioControl Jena GmbH, 07745 Jena, Germany; dominik.driesch@biocontrol-jena.com; 4Central Facility for Microscopy, Helmholtz Centre for Infection Research, 38124 Braunschweig, Germany; manfred.rohde@helmholtz-hzi.de; 5Department of Molecular Bacteriology, Helmholtz Centre for Infection Research, 38124 Braunschweig, Germany; susanne.haeussler@helmholtz-hzi.de

**Keywords:** *Streptococcus pneumoniae*, two-component system 09, carbohydrate metabolism, capsule

## Abstract

*Streptococcus pneumoniae* two-component regulatory systems (TCSs) are important systems that perceive and respond to various host environmental stimuli. In this study, we have explored the role of TCS09 on gene expression and phenotypic alterations in *S. pneumoniae* D39. Our comparative transcriptomic analyses identified 67 differently expressed genes in total. Among those, *agaR* and the *aga* operon involved in galactose metabolism showed the highest changes. Intriguingly, the encapsulated and nonencapsulated *hk09*-mutants showed significant growth defects under nutrient-defined conditions, in particular with galactose as a carbon source. Phenotypic analyses revealed alterations in the morphology of the nonencapsulated *hk09*- and *tcs09*-mutants, whereas the encapsulated *hk09*- and *tcs09*-mutants produced higher amounts of capsule. Interestingly, the encapsulated D39∆*hk09* showed only the opaque colony morphology, while the D39∆*rr09*- and D39∆*tcs09*-mutants had a higher proportion of transparent variants. The phenotypic variations of D39Δ*cps*Δ*hk09* and D39Δ*cps*Δ*tcs09* are in accordance with their higher numbers of outer membrane vesicles, higher sensitivity against Triton X-100 induced autolysis, and lower resistance against oxidative stress. In conclusion, these results indicate the importance of TCS09 for pneumococcal metabolic fitness and resistance against oxidative stress by regulating the carbohydrate metabolism and thereby, most likely indirectly, the cell wall integrity and amount of capsular polysaccharide.

## 1. Introduction

Living cells rely on sensing and interpreting external signals as an adaptive mechanism against changes in their environment such as oxygen, temperature, and pH. This process is known as signal transduction and allows the cell to react immediately and appropriately to specific alterations in their habitat. One of the main signaling mechanisms used by bacteria are the two-component regulatory systems (TCSs). In bacteria, a TCS consists of a transmembrane protein (histidine kinase (HK)) that functions as a sensor and a corresponding cytoplasmic protein (response regulator (RR)) controlling gene expression. The HK can react to external and internal signals by binding a phosphate group to a conserved histidine residue (autophosphorylation) [1]. In general, a histidine kinase is specific to its cognate response regulator; however, crosstalk between different TCSs in the bacterial cell can also occur [2].

For successful colonization of the upper respiratory tract, *Streptococcus pneumoniae* (pneumococci) have evolved TCSs and adapted their use to the host compartments. In pneumococci, 13 histidine kinases and 13 response regulators working in pairs plus one orphan response regulator were identified [3,4,5]. Virulence studies conducted to assess the role of TCSs indicated that 7 TCSs and the orphan response regulator influence bacterial growth and virulence in a mouse respiratory tract infection model [4]. Pneumococcal TCSs are associated with regulation of different processes such as competence and fratricide (TCS02 (WalRK/VicRK/YycFG), 03, 05 (CiaRH), and 12 (ComDE)), bacteriocin production (TCS05, 12, and 13 (BlpRH)), virulence factor expression (TCS02, 03, 05, 06 (CbpRS), 08, 09 (ZmpRS), and 10 (VncRS)), response to antibiotic and cell wall perturbations (TCS02, 03, 05, and 11), environmental stress (TCS01 (SirRH); TCS04 (PnpRS), 05, and 12; and RR14 (RitR)) and nutrients uptake (TCS04, 05, 07, 08, 09, and 12 and RR14) [5,6]. We have shown previously that TCS08 is involved in pneumococcal colonization by regulating the virulence factors PavB, the *rlrA* islet (Pilus-1), and the cellobiose metabolism. Moreover, the TCS08 effect on regulatory processes including colonization is strain-dependent, as shown in a murine model when comparing *S. pneumoniae* strains TIGR4 and D39 [7]. Recently, TCS01 (SirRH) was shown to increase pneumococcal survival in pneumocytes after influenza An infection by transcription of genes involved stress response [6].

TCS09 is among the pneumococcal TCSs whose function and target genes are still unknown. The histidine kinase 09 belongs to the histidine protein kinase class 08 [8], while its cognate RR09 is part of the YesN subgroup containing an HTH_ARAC DNA binding domain [9,10,11]. Initially, it was suggested that TCS09 regulates the expression of the virulence factor ZmpB, but follow up studies were unable to reproduce and confirm this [12,13,14,15]. A previous study suggested that the function of TCS09 is connected to metabolism and virulence [15,16]. Using microarray analysis, it was reported that pneumococcal gene regulation by TCS09 is strain-dependent and that TCS09 plays a role in metabolic processes. Various PTS genes involved in carbon transport were found to be downregulated in an RR09-deficient strain D39, while only three PTS genes (lactose, trehalose, and galactitol) showed a similar effect in TIGR4∆*rr09* [15]. A strain-specific effect of RR09 deficiency was reported with respect to virulence. Strain D39 (serotype 2) pneumococci showed a complete avirulent phenotype in the absence of RR09; however, this phenotype was not observed for a serotype 3 and 4 *S. pneumoniae rr09* knock out mutant in a sepsis and acute pneumonia mouse infection model [16]. The different outcome of loss of function of RR09 in pneumococci is intriguing, but the reasons for the strain specific effects are unknown and require further evaluation. A recent study showed that the switch between opaque (O) and transparent (T) variants of pneumococci is regulated by five response regulators. Pneumococci deficient in RR06, RR09, RR11, or RR14 produced significantly more T than O colonies. Furthermore, it was shown that RR06, RR08, RR09, and RR11, respectively, modify the direction of the DNA inversion reaction in the *hsdS* genes (DNA methyltransferases) catalyzed by the tyrosine recombinase PsrA, which leads to a higher number of opaque variants [17,18]. Thus, TCS09 seems to play an important role in fitness and in phase variation, which could have a significant impact on virulence. In this study, we have explored the role of TCS09 on the pathophysiology of *S. pneumoniae* by transcriptome analysis (RNA-seq) of strain D39 employing mutants deficient for the response regulator (∆*rr09*), the histidine kinase (∆*hk09*), and both TCS09 components (∆*tcs09*). Our data suggest potential target genes for the TCS09 of D39 that are involved in capsule modulation and fine-tuning of the metabolism but not virulence.

## 2. Materials and Methods 

### 2.1. Bacterial Strains and Culture Conditions

Encapsulated and nonencapsulated (Δ*cps*) *Streptococcus pneumoniae* serotype 2 (D39) parental strains and isogenic *tcs09*-mutants (Table 1) were used in this study. Pneumococci grown on blood agar plates (Oxoid, Wesel, Germany) with appropriate antibiotics were inoculated in chemically-defined medium (CDM) RPMI_modi_ (RPMI1640: GE Healthcare; RPMI_modi_: [19]) supplemented with 1% *w/v* glucose or galactose or Todd Hewitt Broth (Roth) supplemented with 0.5% yeast extract and cultivated at 37 °C without agitation up to middle logarithmic phase (OD_600nm_ 0.6). *Escherichia coli* DH5α was used as a host strain for recombinant plasmids for *rr09*, *hk09*, and *tcs09* mutagenesis and cultivated at 37 °C on LB agar or liquid culture with 200 µg/mL erythromycin and 120 rpm agitation.

### 2.2. Molecular Biological Techniques

Chromosomal DNA isolation from pneumococci was performed using phenol-chloroform methodology. In brief, bacteria were cultivated in THY and harvested by centrifugation. The pellet was resuspended in TES buffer and treated with lysozyme, pronase E, RNase A, and N-lauryl sarcosine. Finally, phenol and phenol:chloroform:isoamyl alcohol (25:24:1 ratio) were used to separate the DNA. The resulting DNA was washed with 96% ethanol and stored in Tris-EDTA buffer at −20 °C until further experimentation. DNA regions for mutant generation were amplified by PCR, and the oligonucleotides used (Eurofins MWG, Ebersberg, Germany) are listed in Table 2. PCR products were purified using the Zymoclean Gel DNA Recovery Kit (Zymo Research, Freiburg, Germany), and plasmid DNA (Table 3) was isolated using the Wizard^®^ Plus SV Minipreps DNA Purification System (Promega). The T4 DNA ligase and restriction enzymes were purchased from Thermo Fisher Scientific and New England Biolabs and used according to the manufacturer’s instructions. Transformation of *E. coli* with recombinant plasmids was conducted with calcium chloride competent cells. Transformants were selected on LB agar containing 5 µg/mL erythromycin, and plasmids sequences were verified by DNA sequencing (Eurofins MWG, Ebersberg, Germany).

### 2.3. Generation of Pneumococcal Mutants

Pneumococcal ∆*rr09* (*spd_0574*; SPD_RS03105), ∆*hk09* (*spd_0575*; SPD_RS03110), and ∆*tcs09*, lacking the *rr09* and *hk09* mutants, were constructed by insertion-deletion mutagenesis (Appendix A). Transformation of *S. pneumoniae* D39 using the constructed plasmids was performed in the presence of competence-stimulating peptide CSP1, as described [7]. Transformants were selected on blood agar plates containing 5 µg/mL erythromycin, and all mutants were confirmed on the molecular level by colony PCR and RNA-sequencing.

To generate RR09-deficient pneumococci, the gene *spd_0574* with 600 bp flanking regions upstream and downstream was PCR amplified with primers 1122/1123. The resulting 2049 bp DNA fragment was ligated via TA cloning into vector pGXT [22] and transformed into competent *E. coli* DH5α. To delete the *rr09* gene, the recombinant plasmid pGXT_*rr09* was used as a template for an inverse PCR with primers 1124/1232, which have an incorporated *Hin*dIII restriction site. The PCR product was ligated with the *erm*^R^ cassette amplified with primers 105/106 (pGXT∆*rr09*::*erm*^R^) and transformed into competent *E. coli* DH5α. 

For the construction of the vector pGSP72N∆*hk09*::*erm*^R^, the 3′ flanking region of *spd_0575* was amplified using primers 1127/1128. The erythromycin resistance gene cassette (*erm*^R^) was amplified with primers 1234 and 100. These two DNA fragments were digested with the restriction enzyme *Hin*dIII and ligated, and amplified by PCR with primers 1234/1127. This PCR product was ligated into *Eco*RV linearized vector pGSP72N (this study) and transformed into competent *E. coli* DH5α resulting in pGSP72N_*erm*^R^-3′*hk09*. The 5′ flanking region of the gene *spd_0575* was amplified with primers 1126/1129 and digested with *Nhe*I. This DNA fragment was ligated with *Nhe*I and *Sma*I cleaved vector pGSP72N_*erm*^R^-3′*hk09*, and the ligation product was transformed into competent *E. coli* DH5α cells. 

To produce the mutagenesis plasmid pSP72∆*tcs09*::*erm*^R^, the 5′ flanking region of *spd_0574* and the 3′ flanking region of *spd_0575* were amplified by PCR using primers 1122/1125 and 1128/1127. The obtained PCR fragments were digested with *Nhe*I (5′-*spd_0574*) and *Hin*dIII (3′-*spd_0575*), respectively, and genetically fused with a similarly digested erythromycin resistance gene (*erm*^R^), which was amplified with the primers 1234 and 106. A PCR reaction using the three PCR products with primers 1122 and 1127 was conducted to obtain a mutated *spd_0574* DNA region. This DNA fragment was ligated into vector pSP72 (Promega), which was linearized with *Eco*RV. The genes downstream (or upstream) of the genes knocked-out by the insertion-depletion strategy are continuously expressed, and their expression level is not changed. This is clearly shown in the RNA-seq data (log_2_-fold change RNA expression pattern) (Appendix A).

### 2.4. RNA Purification

Pneumococcal strain D39 and isogenic ∆*rr09*-, ∆*hk09*-, and ∆*tcs09*-mutants were cultivated in glucose supplemented RPMI_modi_ up to an OD_600nm_ of 0.6. Ice cold killing buffer (20 mM Tris HCl pH 7.5, 5 mM MgCl_2_, 20 mM NaN_3_) was added to the bacterial cultures and centrifuged for 10 min at 3200× *g* at 4 °C. The supernatant was removed and the bacterial pellets were immediately frozen in liquid nitrogen and stored at −80 °C. For total RNA extraction, the samples were treated with acidic phenol-chloroform and subjected to TURBO™ DNase (2 U/reaction; Invitrogen, Carlsba, CA, USA) digestion to remove genomic DNA. The RNA was purified using RNA cleanup and concentration kit (Norgen Biotek Corp., Thorold, ON, Canada). The RNA quality was controlled with Agilent 2100 Bioanalyzer (Appendix A), and the amount of RNA was measured using a NanoDrop ND-1000 spectrophotometer (Peqlab). Four replicates were prepared per strain and mutant. 

### 2.5. RNA-Sequencing

Libraries for transcriptomics were generated according to Shishkin et al., 2015 and Bhattacharyya et al., 2019 [23,24]. Briefly, 1 µg of total RNA was fragmented in 150–350 bp, and phosphate groups at the 3′ and 5′ end were removed using FastAP (Thermo Scientific). A further DNA digestion step with the TURBO™ DNase (Invitrogen, Carlsba, CA, USA) was carried out. The fragmented RNA was purified with RNA Clean XP Beads following ligation of phosphorylated barcoded DNA adapters (L01–L32; Appendix A) to the fragmented RNA by a T4 RNA ligase (NEB) with final column-based (RNA clean & concentrator-25, purchased from Zymo Research) purification of the ligated RNA. The rRNA was removed with the RiboZero Kit, and RNA was purified again with the RNA Clean XP Beads. A reverse transcription to cDNA with the two primers AR2 and 3Tr3 (Appendix A) was performed and followed by subsequent removal of the untranscribed RNA using RNA Clean XP Beads. The purified cDNA was enriched by PCR using primers P5 and X01 (Appendix A), while a second DNA adapter was added. The cDNA was then purified again with Clean XP Beads. Sequencing on NovaSeq 6000 (Illumina, San Diego, California, United States) followed in paired-end mode with 50 cycles. Fastq files were assessed for sequence quality using FastQC (version 0.11.5), and an index from the genome from the file NC_008533 (D39) was created with bowtie2 (version 2.3.4.3). The mapping of the reads to the index was also performed with bowtie2. Reads were counted and assigned to the corresponding genome using function featureCounts of R package Rsubread (version 2.2.2). Contrasts were calculated using R package DESeq2 (version 1.28.1). R version 3.4.4 was used. To analyze the differential gene expression of the mutants vs. wild-type (referred as control), DESeq2 (version 1.18.1) was used. A log_2_ fold change threshold was set to 1 and an adjusted p-value cutoff of 0.05. Heatmaps were created using ClustVis [25].

### 2.6. Quantitative Real-Time PCR (qPCR)

Isolated RNA was transcribed in cDNA using Superscript III reverse transcriptase (Thermofisher, Waltham, MA, USA) and random hexamer primers (BioRad, Hercules, CA, USA) according the instructions of the manufacturer. A StepOnePlus thermocycler (Applied Biosystems, Foster City, CA, USA) and the iTaq Universal SYBR Green Supermix (BioRad) were used for quantitative real-time PCR. Enolase (*spd_1012*) was used as a reference gene and to correct for sample variation regarding, for example, reverse transcriptase efficiency. The enolase gene expression was not significantly changed in the mutants, as indicated by RNA-seq (Appendix A). The *agaR* (*spd_0064*), *gadV* (*spd_0066*), *celA* (*spd_0277*), *eng* (*spd_0335*), and *spd_1588* genes were used as targets (Primer list Table 4). The qPCR conditions using 20 ng/µL cDNA as a template were as followed: initial denaturation at 95 °C for 2 min, denaturation at 95 °C for 15 s, primer annealing at 60 °C for 30 s, and extension at 72 °C for 30 s for 40 cycles with a final melting curve step for quality control. Differential gene expression was calculated by the Pfaffl method [26].

### 2.7. Field Emission Scanning Electron Microscopy (FESEM)

Bacteria were cultivated in CDM with glucose as a carbon source until OD_600nm_ 0.6 at 37 °C. After centrifugation, the encapsulated strains were fixed with precooled 2.5% glutardialdehyde, 2% paraformaldehyde, 0.075% ruthenium red, and 75 mM L-lysine acetate salt in cacodylate buffer on ice for 20 min (Lysine-ruthenium-red (LRR) fixation) and washed with cacodylate buffer containing 0.075% ruthenium red. Subsequently, in a second fixation step, the sediment was resuspended with precooled fixing solution (without L-lysine acetate salt) and incubated on ice for 2 h. After washing three times with cacodylate buffer and 0.075% ruthenium red, the samples were dissolved in 1% osmium solution (containing ruthenium red) and incubated at room temperature for 1 h. Finally, the sediment was washed twice with HEPES buffer. Nonencapsulated strains were fixed with 2% glutardialdehyde and 5% paraformaldehyde.

Bacterial aliquots were placed on cover slips, fixed with 1% glutaraldehyde, washed with TE buffer, and dehydrated in a graded series of acetone (10, 30, 50, 70, 90, and 100%) on ice for 10 min for each step. Critical-point drying of the samples was performed with liquid CO_2_ (CPD 30, Bal-Tec, Pfäffikon, Switzerland). Subsequently, the dried samples were covered with palladium-gold film by sputter coating (SCD 500, Bal-Tec) before examination in a field emission scanning electron microscope (Zeiss Merlin, Jena, Germany) using the HESE2 Everhart Thornley SE detector and the in-lens SE detector in a 75:25 ratio at an acceleration voltage of 5 kV.

### 2.8. Transmission Electron Microscopy (TEM)

For TEM, the fixed bacteria were mixed with an equal volume of 2% water agar, solidified, and cut into small cubes. This was followed by dehydration in a graded series of ethanol (10, 30, and 50%) on ice. Subsequently, nonencapsulated strains were incubated in 70% ethanol containing 2% uranyl acetate overnight at 7 °C, whereas LRR fixed samples were incubated in 70% ethanol without uranyl acetate. Afterwards, all samples were dehydrated with 90% and 100% ethanol and infiltrated with aromatic acrylic resin LRWhite (London resin company, London, UK) by applying, firstly, 1 part 100% ethanol and 1 part LRWhite overnight and, secondly, 1 part ethanol and 2 parts LRWhite for 24 h on ice. Pure LRWhite was added with two exchanges within 2 days. Finally, samples were placed in gelatin capsules and filled with pure LRWhite resin. LRWhite resin was polymerized for 2–4 days at 50 °C. Ultrathin sections were cut and counterstained for 3 min with 4% aqueous uranyl acetate. Samples were examined in a Zeiss TEM 910 transmission electron microscope at an acceleration voltage of 80 kV and at calibrated magnifications.

### 2.9. Flow Cytometric Analysis of Capsular Polysaccharide Abundance

The relative abundance of the capsular polysaccharide (CPS) of *S. pneumoniae* D39 and isogenic mutants was measured by flow cytometry, as described [27]. Briefly, pneumococci were grown in CDM with glucose as a carbon source to a final OD_600nm_ 0.6. After sedimentation and resuspension in PBS 2 × 10^8^, CFU pneumococci were incubated for 45 min with specific anti-serotype 2 antibodies (Staten Serum Institute, Copenhagen, Denmark) at 4 °C. Pneumococci were washed with PBS, and bound primary antibodies were labelled with Alexa Fluor 488 conjugated anti-rabbit antibodies (abcam) at 4 °C. Pneumococci were finally fixed with 1% paraformaldehyde, and the fluorescence intensity was analyzed by flow cytometry using a FACS Calibur (BD Biosciences, Heidelberg, Germany). The forward and sideward scatter dot plots were used to detect fluorescent pneumococci. A gating region was set to exclude cell debris and larger bacteria aggregates. At least 50,000 pneumococci (events) were analyzed. The geometric mean fluorescence intensity multiplied by the percentage of bacteria in complex with fluorescent labelled bacteria was calculated and used as a relative value for the capsule amount.

### 2.10. Visualization of Pneumococcal Colony Phase Variation

Pneumococcal strain D39 and isogenic mutants were cultivated in CDM at 37 °C until the mid-logarithmic phase (OD_600nm_ 0.6). Afterwards, 200 CFU per strain and mutant were plated on Tryptic Soy Agar (TSA) plates containing 1,000,000 U catalase. The plates were incubated at 5% CO_2_ at 37 °C for 16 h. The colonies were evaluated for their opacity under oblique transmitted light using a Leica M125 C dissecting microscope and LAS X software.

### 2.11. Autolysis Assay with Triton X-100

The autolysis assay was conducted, as described [28]. In brief, pneumococci grown in CDM were adjusted to an OD_600nm_ of 1 in a solution of PBS in the presence of 0.01% or 0.005% Triton X-100 or in the control reaction, in the absence of Triton X-100. The samples were incubated at 37 °C and 5% CO_2_, and the absorbance (OD_600nm_) was measured every 10 min using a spectrophotometer to quantify cell lysis. 

### 2.12. Hydrogen Peroxide Toxicity Test

Pneumococci were cultivated until the mid-logarithmic phase (OD_600nm_ 0.6) in CDM at 37 °C without agitation. Next, 10 mL cultures (*n* = 3 per strain) were prepared with a final concentration of 0, 5, or 10 mM hydrogen peroxide. The samples were incubated at 37 °C for 30 min. Subsequently, serial dilutions were prepared, plated on blood agar plates, and CFU were counted after 20 h incubation at 37 °C and 5% CO_2_. 

### 2.13. Statistical Analysis

Unless stated otherwise, all the data collected in this study are presented as mean of at least three independent experiments with standard deviation ±SD. The results were statistically evaluated using a two-way ANOVA or the unpaired two-side student’s *t*-test (GraphPad Prism 5.01). A *p*-value < 0.05 was considered as statistically significant.

## 3. Results

### 3.1. HK09 of TCS09 Is Essential for Optimal Growth Of Pneumococci under Defined In Vitro Conditions

To investigate the impact of TCS09 on pneumococcal fitness and physiology in a chemically-defined medium (CDM), we cultivated RR09-, HK09-, and TCS09-deficient pneumococci and their isogenic parental strains D39 and D39∆*cps* in a CDM with glucose or galactose as a carbon source [19]. The growth kinetics of nonencapsulated mutant strains deficient for RR09, HK09, or the complete TCS09 in CDM with glucose as carbon source were similar to the isogenic parental nonencapsulated strain D39∆*cps* (Figure 1). In contrast, the encapsulated mutant D39∆*hk09* showed a significantly (*p* < 0.001) longer generation time (g = 129 min) and final bacterial cell density, as indicated by a lower optical density at 600 nm when compared to the encapsulated wild-type strain D39 (g = 79 min) (Figure 1B). The D39∆*tcs09*-mutant had only a slightly increased generation time (g = 107 min) compared to wild-type D39 (Figure 1C). When we used galactose as a carbon source, encapsulated pneumococci showed an extended lag phase compared to the cultures with glucose as a carbon source (Figure 1G–I). In galactose containing CDM, loss of HK09 substantially decelerated the growth (2.1 fold longer generation time) of encapsulated pneumococci, while the loss of TCS09 had only a minor effect (1.2 fold) (Figure 1H,I and Table 5). Interestingly, the nonencapsulated *hk09*- and *tcs09*-mutants were not able to grow in galactose supplemented CDM (Figure 1K,L and Table 5). Thus, under heterofermentative growth conditions, the deficiency of HK09 or the complete TCS09 in the encapsulated strain significantly decelerate growth (Figure 1H,I) and lead to a loss of growth in the nonencapsulated strains. Additionally, encapsulated and nonencapsulated D39 parental strains and their isogenic *tcs09*-mutants were cultured in THY, where no growth defects of the *tcs09*-mutants were observed (data not shown). Taken together, loss of HK09 or the complete TCS09 in *S. pneumoniae* significantly affects pneumococcal growth kinetics depending on the carbon source available under nutrient defined conditions.

### 3.2. Analysis of the D39 Transcriptome under In Vitro Conditions

To assess the effects of the TCS09 components on the transcriptome level under nutrient defined conditions, we cultured the encapsulated *S. pneumoniae* D39 and isogenic mutants in glucose supplemented CDM and isolated total RNA from pneumococci collected in the exponential growth phase (OD_600nm_ 0.6). We performed RNA-seq analysis using four biological replicates per strain and assessed the overall variance in gene expression by means of Principal Component Analysis (PCA) (Figure 2A). Interestingly, only the samples obtained from D39∆*hk09* were clearly separated from those of the wild-type D39 and other mutants. The respective biological replicates grouped together, except for the D39∆*tcs09* samples, where two replicates clustered together with the wild-type and ∆*rr09* samples, and two replicates were separated from all other samples. Our further analysis revealed that in the two deviant samples of D39∆*tcs09*, especially genes belonging to the competence cluster of *S. pneumoniae*, were highly upregulated (Appendix A and Appendix A). 

For the analysis of differential gene expression in the mutants compared to the wild-type log_2_ expression levels < −1 and > 1 and an adjusted *p*-value < 0.05 were considered significant. Because we observed a heterogenous transcriptome between the replicates of D39Δ*tcs09*, the coefficient of variation (CV) values of the individual genes in all strains were determined for all strains, and genes with values > 1.5 x interquartile range (IQR) were not considered for further analysis and are displayed in the heat maps with black asterisks (Figure 2B, Appendix A). According to the specified criteria, a total of 67 protein encoding genes showed significant differences in expression between one of the three mutants and the D39 wild-type. Four genes in D39Δ*rr09*, 67 genes in D39Δ*hk09*, and 0 genes in D39Δ*tcs09* were differentially regulated compared to the wild-type D39 (Figure 2B, Appendix A and Appendix A). Most of these genes encode proteins with functions in intermediary metabolism (18 genes), environmental information processing (12 genes), and genetic information processing (8 genes) (Figure 2B and Appendix A). The highest change in mRNA level within the category of transporters was detected for the PTS encoding *aga* operon *gadVWEF* (galactosamine-specific PTSBCDA component), which was upregulated in the *rr09*- (fold change: 2.2–3.4) and *hk09*-mutants (fold change: 20.2–38.6). In addition, *bgaC*, *agaS*, and *galM*, all involved in galactose metabolism and located in the same operon as *gadVWEF*, exhibited similarly increased mRNA levels in the *rr09*- and *hk09*-mutants as *gadVWEF*. Notably, in the HK09 deficient encapsulated D39 mutant, the mRNA level of *agaR* (*spd_0064*), which is a transcriptional regulator of the GntR family and acts as repressor of the *aga* operon [29], was strongly downregulated (fold change: 17.5). AgaR, also referred to as CpsR, is also suggested to be involved in capsule expression regulation [30]. The only major virulence factors contributing to host-pathogen interactions we found to be significantly upregulated on the mRNA level in the *tcs09*-mutants of D39 were *pspA* (*spd_0126*) and *phtD* (*spd_0889*) in D39Δ*hk09*. Moreover, we identified a number of genes coding for proteins with unknown function that were differentially expressed (Figure 2B, Appendix A and Appendix A).

### 3.3. HK09 Regulates Capsule Expression and Carbohydrate Metabolism

To study the effects of *tcs09*-deletion on the pneumococcal genes involved in metabolic processes, capsule regulation, and competence, we used qPCR to validate the findings of the RNA-seq analysis. For qPCR, we repeated the isolation of RNA from D39 wild-type bacteria and isogenic *tcs09*-mutants grown in CDM with glucose as a carbon source. For the Δ*rr09*-mutant, we observed a 2.3 fold increase of mRNA expression of *spd_1588*, a hypothetical protein, whereas *agaR*, *gadV* (galactose metabolism), *celA* (cellobiose metabolism), and *eng* (unclassified carbohydrate metabolism) expression was unaffected compared to the wild-type (Figure 3A). In D39Δ*hk09*, we determined significant upregulation of *gadV* (44.1 fold), *eng* (93.4 fold), and *spd_1588* (23.5 fold) and a significant downregulation of *agaR* (8.8 fold) (Figure 3A). These results are in agreement with our RNA-seq analysis, which showed similar fold changes. Additionally, qPCR with the Δ*tcs09*-mutant revealed higher expression of *spd_1588* (2.2 fold) (Figure 3A). Because the RNA-seq data showed an inconsistent gene expression of the competence cluster in D39Δ*tcs09,* we analyzed the expression of four competence genes by qPCR. Importantly, the qPCR results indicated no significant effect of TCS09 deletion in the regulation of the competence genes (Figure 3B).

### 3.4. Phenotypic Characterization of TCS09-Deficient Pneumococci

Expression of *agaR* was significantly downregulated in the *hk09*-mutant, and mutants deficient for HK09 showed an altered growth behavior (Figure 1) as well as an upregulation of the genes involved in metabolic processes (Figure 2 and Figure 3). We therefore analyzed our set of parental D39 strains and isogenic mutants cultured to mid-exponential phase for changes in the cell morphology and amount of CPS. Our field emission scanning electron microscopy (FESEM) suggested differences in the surface complexity of the capsular polysaccharide layer of D39Δ*rr09* and D39Δ*tcs09*. The Δ*rr09*- and Δ*tcs09*-mutants consist of two different populations showing a high variation in their roughness phenotype. Interestingly, the *hk09*-mutant presented a thicker CPS compared to its parental strain and cognate mutants (Figure 4A). As a control, the nonencapsulated D39 mutant was used, which showed a clear surface roughness in comparison to its isogenic encapsulated wild-type strain D39 (Figure 4A). In the nonencapsulated strains, a high number of cell ends of D39Δ*cps*Δ*hk09* and D39Δ*cps*Δ*tcs09* looked swollen and wrinkly (Figure 4A). The cellular surface of the swollen ends had a morphology that was clearly distinct from that of the smooth surface of wild-type or RR09-deficient D39Δ*cps* pneumococci (Figure 4A). Additionally, small outer membrane vesicles (white arrows), probably due to cell wall degradation or alterations in peptidoglycan, were detectable on the surface of all strains with a higher proportion for the *hk09*- and *tcs09*-mutants. 

In addition, we applied transmission electron microscopy (TEM) to compare cell morphology, amount of CPS, and intracellular appearance of the *tcs09*-mutants with their D39 wild-types. Our ultrathin sections of *S. pneumoniae* D39Δ*rr09* suggested a detachment of CPS and a lower capsule density on the cell surface. In contrast, D39Δ*tcs09* presented a cell population with a variation in the amount of surface CPS (Figure 4B). This phenomenon was not detectable for D39Δ*hk09*; instead, this strain exhibited a thicker capsule layer than the wild-type or its cognate mutants (Figure 4B). Further analysis of the *rr09*-mutant of D39 revealed a less dense cytoplasm, probably due to less proteins and ribosome accumulation (Figure 4B). Additionally, the cytoplasm of all *tcs09*-mutants of the nonencapsulated strains exhibit some distinct white small areas in the bacterial cytoplasm, but these seemed not to be vesicles due to a missing membrane, and no contact to the cytoplasmic membrane is detectable (Figure 4B). All pneumococcal membranes and cell walls appeared to be intact, indicating that complete deletion of the TCS09 system did not have a deleterious effect on the cell surface structure of the *S. pneumoniae* strains (Figure 4B).

### 3.5. Impact of TCS09 on the CPS Amount on the Pneumococcal Surface

Our CPS illustration of pneumococcal *tcs09* regulatory mutants by FESEM and TEM suggested different amounts of CPS compared to the isogenic D39 wild-type. Therefore, we assessed and quantified the relative amount of CPS linked to the pneumococcal surface by flow-cytometry (Figure 5). The CPS of D39 was detected using a serotype 2 antiserum and secondary Alexa conjugated antibody. The fluorescence intensity of the labeled pneumococci was quantified by flow cytometry. As control, the nonencapsulated D39Δ*cps* strain was used, and D39Δ*cps* revealed a significantly lower fluorescence signal compared to the encapsulated D39 strain (Figure 5). Our measurements indicated that a high proportion (99.7% of D39, 91.3% of D39Δ*rr09*, 99.8% of D39Δ*hk09*, and 98.6% of D39Δ*tcs09*) of pneumococci were fluorescent and, hence, were positive for CPS (Appendix A). Importantly, the histograms (Figure 5A) and the relative quantification of CPS (Figure 5B) indicated that pneumococci deficient in HK09 or the complete TCS09 had, at least proportionally, a significantly higher capsule amount compared to the wild-type D39. Interestingly, the *rr09*- and *tcs09*-mutants showed heterogeneous populations regarding the CPS expression. A minor proportion shows low levels of fluorescence intensity, indicating low amounts of CPS, while a higher proportion showed a high fluorescence intensity, indicating a high amount of CPS (Figure 5). Immediately at time point 0, the population of the *rr09*-mutant could be divided into a lower capsule expressing population and a wild-type like population (Figure 5 and Appendix A). For the *tcs09*-mutant, which is deficient in RR09 and HK09, a similar effect was observed, but it was less pronounced compared to D39Δ*rr09* (Figure 5A and Appendix A). Furthermore, capsule detachment was investigated in mutants resuspended in PBS over a period of 60 min. The flow cytometric analyses showed that the proportion of low CPS expressing pneumococci of D39Δ*rr09* and D39Δ*tcs09* remained constant, suggesting that the reduced amount of CPS in these strains is not caused by CPS shearing during sample preparation (Figure 5B). Taken together, our relative quantification of the CPS confirmed the electron microscopic data and indicates that TCS09 contributes to pneumococcal CPS expression.

### 3.6. Impact of TCS09 on Pneumococcal Phase Variation

We further analyzed the mutants for their change of colony opacity phenotype. *S. pneumoniae* D39 and isogenic mutants were plated out on TSA plates supplemented with catalase, and single colonies were observed under oblique transmitted light. Opaque colonies are typically of convex elevation compared to transparent colonies, which have a central dent, as shown in Figure 6A. Most of the D39Δ*rr09* and D39Δ*tcs09* colonies appeared smaller and transparent (74% and 65%, respectively) compared to the opaque wild-type D39 colonies (Figure 6). Interestingly, the D39Δ*hk09* mutant showed the same opaque phenotype as the isogenic wild-type strain D39 (Figure 6). These observations are in agreement with our findings regarding the CPS amount, as indicated by electron microscopy and flow cytometry.

### 3.7. Autolysis Assay with Triton X-100

Our FESEM studies suggested alterations in the cell morphology and cell wall integrity in D39Δ*cps*Δ*hk09* and D39Δ*cps*Δt*cs09* (Figure 4A). Thus, we hypothesized that these bacterial cells lyse faster under stress conditions. To accelerate or induce pneumococcal cell lysis, Triton X-100 was used at a low concentration to study the impact of TCS09 on pneumococcal stability and autolysis. Triton X-100 is a non-ionic detergent with a polar head group disrupting the hydrogen bonds in lipid bilayers. The integrity of the lipid membrane is disrupted, which ultimately leads to cell lysis. Pneumococci were cultured in CDM until OD_600nm_ 0.6 and resuspended in PBS containing Triton X-100 at a final concentration of 0.005% or 0.01%. Control samples were incubated in PBS without Triton X-100. The results show the survival rate given as a normalized percentage (%) by taking the initial OD at time 0 as 100% (Figure 7). In general, we monitored that the nonencapsulated D39 strain lysed faster than the encapsulated strains within the same time period (Figure 7). The autolysis assay with D39Δ*cps* and its isogenic *tcs09*-mutants showed that the HK09- and the TCS09-deficient mutants exhibited an increased Triton X-100 induced autolysis rate compared to the parental strain D39Δ*cps*. After 50 min, 50% of D39Δ*cps*Δ*hk09* and D39Δ*cps*Δ*tcs09* showed lysis, while for D39Δ*cps* this was the case only after 70 min. In the same period of time, only 15% of the RR09-deficient pneumococci presented lysis (Figure 7A). A significant dose-dependent effect was not observed. When we tested the lysis of the encapsulated strains, D39Δ*tcs09* exhibited an autolysis rate of 50% in 1 h, when pneumococci are exposed to 0.01% Triton X-100, which is significantly different to the wild-type and other mutants showing an autolysis rate of 20%. After 80 min and in the presence of 0.01% Triton X-100, the Δ*rr09*-mutant of D39 started to lyse faster than the wild-type D39 (Figure 7B). In conclusion, a higher autolysis rate compared to isogenic parental strains is indicated for D39Δ*cps*Δ*hk09*, D39Δ*cps*Δ*tcs09*, D39Δ*tcs09*, and partially for D39Δ*rr09*.

### 3.8. Impact of TCS09 on Resistance against Oxidative Stress

To assess the sensitivity of *tcs09*-mutants against oxidative stress, we investigated pneumococcal survival in the presence of exogenous hydrogen peroxide resembling oxidative stress conditions. Reactive oxygen species (ROS), such as hydrogen peroxide, can diffuse in a limited manner through the lipid bilayer of a cell membrane and cause oxidation of DNA, proteins, and membrane lipids, resulting in the accumulation of irreversible oxidative damages and cell lysis [31,32,33,34]. Because *hk09*- and *tcs09*-mutants showed alterations of the cell wall (Figure 4A), hydrogen peroxide might diffuse easier into the cell and cause cell damage and lysis. D39, D39Δ*cps*, and its isogenic *tcs09*-mutants were cultivated in CDM to an OD_600nm_ of 0.6, and oxidative stress conditions were created by adding different hydrogen peroxide concentrations (5 and 10 mM). No significant difference between the encapsulated wild-type and *tcs09*-mutant strains were observed when exposed to different concentrations of hydrogen peroxide (Figure 8A). However, we detected a significant difference between D39Δ*cps* and the isogenic D39Δ*cps*Δ*hk09* as well as with D39Δ*cps*Δ*tcs09* mutants, with a dose-dependent lower survival rate for the mutants. In contrast, the RR09-deficient mutant D39Δ*cps*Δ*rr09* showed a similar resistance to oxidative stress conditions induced by hydrogen peroxide as the parental strain D39Δ*cps* (Figure 8B). These data are in accordance to the higher sensitivity of the D39Δ*cps*Δ*hk09* and D39Δ*cps*Δ*tcs09* mutants for Triton X-100.

## 4. Discussion

Pneumococci are extracellular human opportunistic bacteria colonizing, mostly symptomless, the upper respiratory tract. However, these pathobionts are also capable of causing severe lung infections, septicemia, and meningitis [35,36,37]. The host compartments encountered by pneumococci during their extracellular fate differ significantly with respect to nutrient availability, physiological conditions, and immune defense factors. Bacteria sense their environment via TCS, and the induced outside-inside signaling allows the adaptation to specific physiological conditions via altered gene expression. This in turn endows the bacterium, e.g., with a higher abundance of specific fitness factors or a modified set of virulence factors that enable bacterial growth and survival in various ecological host niches [12,27,38,39,40]. Among the set of pneumococcal TCS, TCS09 has been previously suggested to be involved in fitness adaptation. TCS09 is also referred to as ZmpRS because initial studies suggested a regulation of the virulence factor ZmpB, which could not be confirmed in follow up studies [12,13,15,16]. Further studies showed that TCS09 regulates in a strain-specific manner pneumococcal virulence and metabolic processes. However, these studies have mostly compared the wild-type with the RR09 deficient isogenic mutant [15,16,17]. 

Initially, we compared the growth and transcriptomes of *S. pneumoniae* serotype 2 strain D39 mutants deficient for RR09, HK09, or the complete TCS09 with the isogenic wild-type D39 to comprehensively characterize the physiological role of the pneumococcal TCS09. Our growth experiments under nutrient defined conditions in a CDM with glucose as the sole carbon source showed similar growth kinetics and generation time for wild-type D39 and its isogenic mutants D39Δ*rr09* and D39Δ*tcs09*. The generation time of the nonencapsulated D39Δ*cps* and its isogenic mutants was also similar, albeit the overall growth was slower compared to the wild-type D39 (Figure 1A–F). In contrast, a significant decelerated growth was monitored for the D39Δ*hk09* mutant compared to the D39 wild-type strain (Figure 1B). The growth defects of mutants were even more pronounced when we used galactose as a carbon source (Figure 1G–L). The loss of function of HK09 and both TCS09 components resulted in a significant growth defect when compared to the wild-type D39. When we tested the nonencapsulated mutants, the deficiency of HK09 and the complete TCS09 regulatory unit impaired growth in the CDM with galactose as a carbon source (Figure 1K,L). Interestingly, the generation time of the nonencapsulated D39Δ*cps* is shorter compared to the encapsulated wild-type in CDM with galactose as a carbon source. Pneumococci are strictly fermentative human pathogens relying on carbohydrate metabolism to generate energy for growth [41,42,43,44,45]. When we compared the growth kinetics of encapsulated pneumococci in the context of the carbon source used, galactose slowed down pneumococcal growth but leads to a higher optical density, at least in D39 and the isogenic *rr09*-mutant. Notable, pneumococci ferment glucose homofermentatively with the main products lactate and acetate, while the fermentation with galactose is heterofermentative with mixed-acid products, and lactate is not produced [42,46,47]. Our results with a lower growth rate with galactose as a carbon source are in accordance with a previous study, which further showed the importance of the lactate dehydrogenase for the pyruvate metabolism of pneumococci [47]. Galactose is a constituent of mucin in the human nasopharynx and is imported by several ATP binding cassette (ABC) transporters and PTSs (*gadVWEF*, *spd_0088*–*spd_0090*, *manNML*, and *spd_0559*–*spd_0561*). Intracellular galactose or galactoseamine is metabolized by the tagatose pathway. The genes encoding enzymes of this pathway (*lacABCD, lacT*, *lacF-2*, *lacE-2*, and *lacG-2*) are higher expressed in the presence of galactose [46,48,49]. It has to be mentioned that *lacD* in D39v is annotated as a pseudogene according to PneumoBrowse (https://veeninglab.com/pneumobrowse, accessed on 16 November 2020) [49]. Additionally, galactose can be further metabolized in the Leloir pathway (*galM*, *galE-1*, *galE-2*, *galT-2* and *galK*) [41,42].

The observed growth defects correlated with our RNA-seq based transcriptome analysis. We successfully identified differential expression of 67 protein encoding genes when wild-type and mutant pneumococci were cultivated in glucose-containing CDM. The most significantly regulated genes were found in the mutant D39Δ*hk09* (Figure 2B). Comparisons of the transcriptome data (Figure 2B) and qPCR data (Figure 3A) between the D39 wild-type and Δ*hk09*-mutant revealed in each an elevated expression of genes involved in carbohydrate metabolism, especially of genes contributing to galactose metabolism (*spd_0065–spd_0071*). As mentioned above, these genes are organized in a single transcriptional unit, the *aga* operon (Figure 2C). This operon contains an exo-β-galactosidase (*bgaC*), the PTS *gadVWEF* involved in galactoseamine transport, a sugar isomerase (*agaS*), and an aldose 1-epimerase (*galM*) [29]. According to the study by Afzal et al., the promotor region of this operon exhibits an *agaR* binding motif (5′-ATAATTAATATAACAACAAA-3′). In an *agaR*-mutant, the *aga*-operon was upregulated, whereas no other operons were significantly changed [29]. Intriguingly, one of the significantly downregulated genes (17.5-fold) under our growth conditions in D39Δ*hk09* was *agaR* (*spd_0064*) (Figure 2B). Regulation of *agaR* was verified by qPCR, showing an 8.8-fold down-regulation (Figure 3A). The upregulation of the *aga* operon in the *hk09*-mutant fits perfectly with the downregulation of *agaR*, which acts as a repressor of the operon (Figure 2B and Figure 3A). AgaR is a GntR family regulator, which is further suggested to be involved in capsular polysaccharide biosynthesis. AgaR (referred to as CpsR in [29]) was described to interact directly with the *cps* promotor and to control negatively the *cps* gene transcription [30]. However, our transcriptome data did not support this finding, because we could not measure a lower expression of the *cps*-operon. BgaC, a non-classical surface protein, has a sugar specific hydrolysis activity for Gal*β*1-3-GlcNAc moiety of oligosaccharides, a known receptor for pneumococcal attachment [50,51]. The cleavage releases galactose that can now be imported and used in further metabolic processes. A deficiency of BgaC enhances pneumococcal adherence to epithelial cells and therefore it can be responsible for the release of *S. pneumoniae* from the host cell surface, so that pneumococci can now perform transcellular migration [51]. 

Two replicates of the D39Δ*tcs09* mutant showed increased expression of competence genes in the RNA-seq analysis, most probably indicating higher extracellular levels of the competence-stimulating peptide (CSP) and concomitant activation of the ComE and ComX regulons. Because of the almost identical growth of the four replicates, this observation was hard to explain but did not seem to be directly linked to TCS09 deficiency. Accordingly, qPCR-based analysis of independently generated samples confirmed that competence gene expression in D39Δ*tcs09* cultures is consistent and comparable to the D39 wild-type.

For D39∆*rr09*, we measured only a significant upregulation for the *gadVW and agaS* gene expression, which could not be confirmed by qPCR (Figure 2B and Figure 3). In accordance with the qPCR, the *rr09*-mutant did not show any regulation of *agaR*, which suggests a direct role of RR09 on transcription of this operon (Figure 2B). An earlier performed microarray-based transcriptome analysis with D39Δ*rr09* cultivated in complex media such as Todd Hewitt Broth supplement with yeast (THY) or brain-heart infusion (BHI) demonstrated a significant down-regulation of *gadVWEF* gene expression [15]. However, because these results are not in accordance with our findings, we hypothesize that the main reasons for these contradicting results are the use of different culture media. Our RNA-seq based transcriptome analyses were done with pneumococci cultured in CDM containing a defined carbon source and mix of nutrients, while lacking peptides, lipids, polymers, and growth factors. In CDM cultured pneumococci, especially genes in amino acid metabolism, are upregulated compared to pneumococcal THY cultures [43,44,52]. Both, THY and BHI, are complex media, and pneumococcal growth is faster and relies, in addition to the carbon source, on the uptake of the oligopeptides present in these media [27,42,43,44,52]. Thus, the selection of the culture medium or especially the availability of nutrients in the host can influence the gene regulation. These findings are not surprising considering the versatile adaptation of pneumococci to its different host compartments during colonization, dissemination in the blood, or other invasive infections such as meningitis [27,38,39,52]. 

Although we could not demonstrate a direct role of RR09 on expression of virulence factors, another previous study described an avirulent phenotype of the D39Δ*rr09* mutant in murine models of intraperitoneal, intranasal, and intravenous infection [16]. The reasons for the observed attenuation are still unknown. Because a mutant with *gadE* knockout showed no difference in virulence compared to the wild-type [15], the *aga* operon does not play a direct role for the diminished virulence of *S. pneumoniae* D39. Whether this attenuation is associated with the regulatory effect of RR09 on the tyrosine recombinase PsrA-catalyzed DNA inversion reactions that leads to a higher proportion of opaque colonies remains open [17,18]. 

However, our data suggest a potentially indirect role of TCS09 in pneumococcal virulence. The mutant deficient of HK09 showed an upregulation of *pspA* (2.6-fold) and *phtD* (2.1-fold) (Figure 2B). PspA is involved in immune defense by inhibiting complement activation and colonization due to binding and inactivation of the bactericidal apo-lactoferrin [53,54,55]. *S. pneumoniae* encodes four Pht proteins: PhtA, PhtB, PhtD, and PhtE, which are important in zinc acquisition and play a role in pneumococcal adhesion in a strain dependent manner [56,57]. The role of TCS09 and its HK and RR09 has to be re-investigated in different experimental mouse models using pneumococcal strains with different genetic backgrounds. This is important because the regulatory mechanisms and consequences for virulence seem to be variable among pneumococcal strains [19]. 

Our transcriptome data indicated changes of metabolic gene expression, particularly in the *hk09*-mutant, and we measured growth defects of our mutants D39Δ*hk09,* D39Δ*cps*Δ*hk09*, and D39Δ*tcs09*. This prompted us to elucidate whether these changes are associated with phenotypic and morphologic alterations. We therefore assessed the morphology of encapsulated and nonencapsulated D39 and we further illustrated the capsular polysaccharide in CPS expressing D39 strains and mutants by electron microscopy.

Our FESEM and TEM indicated three phenotypic differences between the parental strains and the mutants. These was (i) a significantly higher amount of CPS in D39Δ*hk09* or in a subpopulation of D39∆*tcs09*, (ii) the production of outer membrane vesicles, which are in particular visible in the nonencapsulated *hk09*- and *tcs09*-mutants*,* and (iii) a wrinkly and swollen surface of the nonencapsulated *hk09*- and *tcs09*-mutants (Figure 4A,B). The quantitatively higher amount of CPS in D39Δ*hk09* and D39∆*tcs09* was confirmed by flow cytometry (Figure 5), and we hypothesize, therefore, that the growth defect in CDM is due to the energy consumption of the CPS production. This effect is probably less pronounced in the D39Δ*tcs09* mutant because our phenotypic analysis indicated that the planktonic culture of the mutant D39Δ*tcs09* in CDM with glucose consists of a mixture of low-encapsulated and high-encapsulated pneumococci (Figure 5 and Appendix A). The production of pneumococcal CPS under nutrient-defined conditions, as in the case of growth in CDM, causes a competition for energy against central carbon metabolism [58]. In 2011, Carvalho et al. analyzed the amount of capsule in *S. pneumoniae* D39, which showed a higher production in galactose-containing medium compared to glucose-containing medium [41]. These finding are in agreement with our data showing an even more pronounced growth defect of D39Δ*hk09* and D39Δ*tcs09* in CDM with galactose as carbon source compared to growth in CDM with glucose (Figure 1H,I). So far, we cannot explain the dramatic growth defect of nonencapsulated mutants deficient in HK09 or TCS09 (Figure 1K,L). 

The variation in CPS amount has also been described when pneumococci alter their opacity [59], and the contribution of RR09 to the switch from the transparent (T) to the opaque (O) phenotype has also been described [17,18]. Furthermore, a Δ*bgaC* mutant (*bgaC*: first gene of the *aga* operon) in strain ST606 (derivative of ST556, serotype 19F) showed more T colonies than O colonies, which indicates that the transcription of the *aga* operon is involved in the stabilization of O variants [17]. Our data strongly support these previous studies because the upregulation of the *aga* operon in D39Δ*hk09* is associated with higher amounts of CPS and with exclusively O colonies. Contrary, the deficiency of RR09 or complete TCS09 resulted in a mixture of O and T variants (Figure 6). 

The nonencapsulated *hk09*- and *tcs09*-mutants showed additional phenotypes: the production of membrane vesicles and a wrinkly and swollen surface (Figure 4A,B). Bacterial outer membrane vesicles (OMV) consist of various types of lipids, membrane proteins, DNA, and RNA [60,61]. In *S. pneumoniae*, these vesicles are enriched for lipoproteins, short-chain fatty acids, and pneumolysin [62]. Membrane vesicles contribute to the digestion of nutrients, quorum sensing signaling, bacterial defense mechanism by binding antimicrobial peptides, and toxin transport [63,64,65,66]. The Gram-positive pneumococci produce vesicles actively from the plasma membrane [62]. For *B. subtilis* and *P. aeruginosa*, it is known that endolysin induced peptidoglycan lysis (explosive cell lysis) is involved in OMV production of bacteria in a passive manner [67,68]. Membrane vesicle formation is further possible via cell lysis that can be triggered by autolysins. Sle1, an autolysin in *S. aureus*, modulates the membrane vesicle production by altering the cell wall permeability and strengthens this hypothesis [69].

Our phenotypic analysis by FESEM and TEM suggests that the cell membrane and cell wall of nonencapsulated *hk09*- and *tcs09*-mutants are instable and that membrane vesicle formation is a side effect of cell lysis and does not happen actively (Figure 4A). Because the D39Δ*cps*Δ*rr09* showed almost no membrane vesicles or alterations of the cell morphology, we hypothesized that this mutant is similar to the parental strain less susceptible to cell lysis compared to the *hk09*- and *tcs09*-mutants. To test this hypothesis, we induced pneumococcal autolysis Triton X-100 treatment (Figure 7). Indeed, the deficiency of HK09 and the entire TCS09 enhanced autolysis in the nonencapsulated mutants, whereas the *rr09*-mutant is even more resistant to lysis than the wild-type (Figure 7A). These data suggest a correlation between the formation of OMVs and an increased sensitivity to cell lysis. For the encapsulated D39 and isogenic mutants, the results are different. The encapsulated Δ*tcs09*-mutant showed a fast autolysis rate within 40 min, followed by stabilization to a specific survival rate. In contrast, the *rr09*-mutant was initially as stable as the wild-type but started to lyse faster after 60 min of Triton X-100 treatment (Figure 7B). Together, the data suggest a decisive role of CPS for sensitivity to cell lysis. The inhibitory effect of CPS on pneumococcal autolysis has already been shown; e.g., the nonencapsulated R6 showed more lysis in comparison to the encapsulated D39 strain. Similarly, the deletion of the *cps* locus in the D39 strain resulted in an increased lysis of this mutant. It is hypothesized that the capsule is able to block the access of the autolysins such as LytA to its target structure peptidoglycan or slows down the translocation of LytA to the cell wall [70]. 

The encapsulated D39Δ*rr09* and D39Δ*tcs09* showed signs for capsule detachment in our TEM (Figure 4B), and two distinct populations (one with less capsule, one with more capsule) were identified in the flow cytometric-based quantification of CPS (Figure 5A and Appendix A). In addition, a higher number of transparent colonies (low capsule) were observed (Figure 6) in these mutants, which showed a higher tendency for cell lysis. Thus, in *tcs09*-mutants, in which the amount of CPS is reduced or CPS is detached, cell lysis is propagated, while in the nonencapsulated mutants we showed a correlation between OMV formation and cell lysis. 

A question that was still open at this point was whether the loss of CPS or the formation of OMV influences pneumococcal survival during stress. Our oxidative stress assay with hydrogen peroxide (H_2_O_2_) as a reactive oxygen species (ROS) indicated a lower survival rate of nonencapsulated *hk09*- and *tcs09*-mutants compared to the parental strain D39Δ*cps* or D39Δ*rr09* (Figure 8). These data are in agreement with our hypothesis that a higher proportion of OMVs correlates with a reduced integrity of the cell wall and, hence, higher sensitivity against stress conditions. 

The mild impact of TCS09 on gene expression alterations suggests a role for TCS09 as a fine tuning regulatory system. There are several conceivable reasons for this observation: (i) the RR09 is modulating additional regulatory changes in the absence of the HK09 such as repression of transcription in its non-phosphorylated state, and (ii) cross talk among the different pneumococcal TCSs may provide an explanation as to why we see effects mostly only in D39Δ*hk09* and not all three *tcs09*-mutants.

## 5. Conclusions

Taken together, TCS09 is most likely not directly involved in the regulation of CPS genes or virulence factors. Instead, the phenotypic and morphologic changes observed are indirectly induced by changing the carbohydrate metabolism and thereby the cell wall integrity and amount of CPS. This in turn has significant consequences for metabolic fitness and resistance against stress. Therefore, it is plausible that TCS09 is most likely also essential for full virulence of pneumococci under infection conditions. However, this has to be confirmed in further studies. 

## Figures and Tables

**Figure 1 microorganisms-09-00468-f001:**
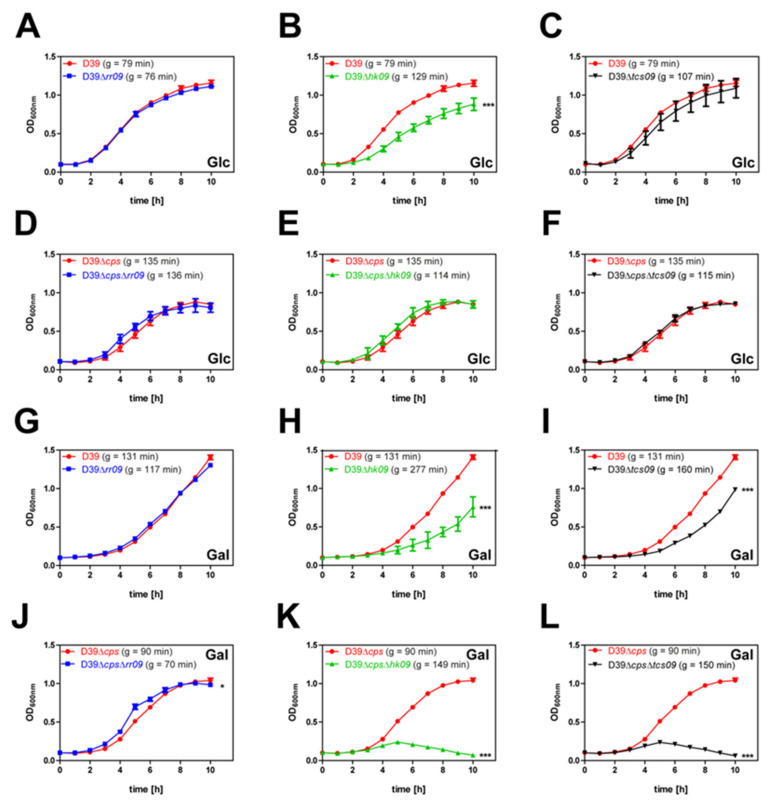
Growth kinetics of RR09-, HK09-, and TCS09-deficient D39 pneumococci and parental strains under defined nutrient resources. The parental strains D39 or D39Δ*cps* and their isogenic Δ*rr09*-, Δ*hk09*-, and Δ*tcs09*-mutants were cultivated in a chemically-defined medium (CDM) with glucose or galactose as a carbon source at 37 °C under microaerophilic conditions. (**A**–**F**) Growth behavior with glucose as a carbon source. (**G**–**L**) Growth kinetics with galactose as the sole carbon source. Results are presented as the mean ± SD for three independent experiments. The mean value of the generation time of the respective strain is given in brackets. A two-way ANOVA proved a significance with *p*-value * < 0.05 and *** < 0.001 relative to the parental pneumococcal strain. g: generation time.

**Figure 2 microorganisms-09-00468-f002:**
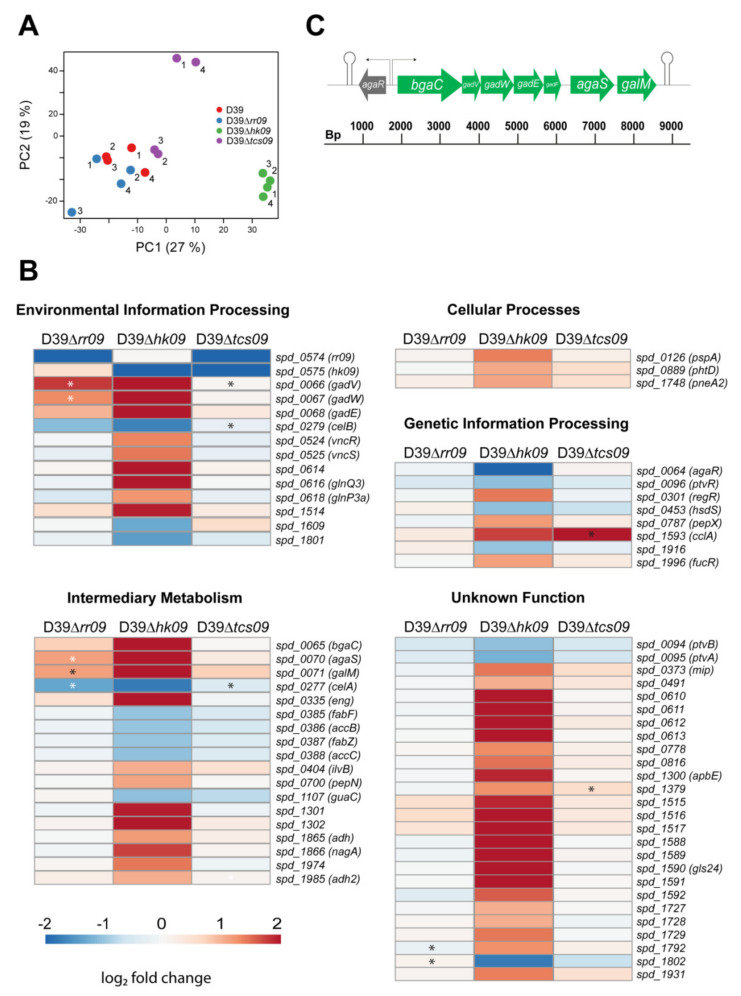
Expression profile of two-component regulatory system 09 mutants. (**A**) Principal component analysis (PCA) of all *S. pneumoniae* samples with color codes. (**B**) Heat maps showing genes with altered expression compared to the wild-type strain D39. Red boxes show upregulation and blue boxes show downregulation in gene expression. The heat maps are divided into different functional categories. The boxes represent the average log_2_ fold change values of four replicates of each strain. A black asterisk (*) means that the coefficient of variation (CV) values of the corresponding gene in the strain replicates is > 1.5 x interquartile range (IQR). White stars indicate a significantly different expression pattern. All presented genes in D39Δ*hk09* are significantly changed. Heat maps showing the log_2_ fold changes in individual replicates of D39Δ*rr09*, D39Δ*hk09*, and D39Δ*tcs09* are depicted in Appendix A. (**C**) Genomic organization of the *aga* operon in *S. pneumoniae* D39. Genes within the *aga* operon are shown in green, and the gene encoding AgaR is upstream of the *aga* operon and shown in grey. The large and filled arrows represent their relative gene size and orientation in the genome. Transcriptional start sites are indicated with thin arrows and terminators with lollypops. Schematic representation was designed according to information from the Pneumobrowser (https://veeninglab.com/pneumobrowse visited on 19 September 2020).

**Figure 3 microorganisms-09-00468-f003:**
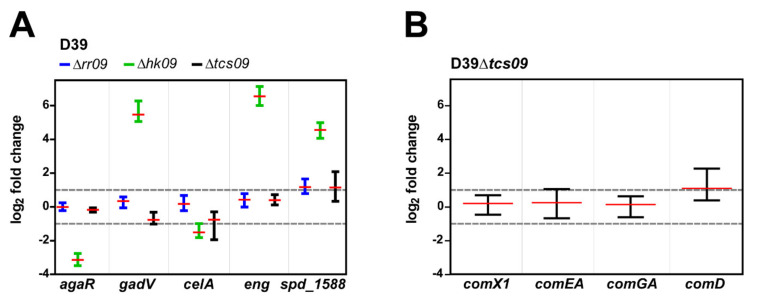
Differential gene expression in *S. pneumoniae* D39 and isogenic *tcs09*-mutants using qPCR. Pneumococci were grown in CDM with glucose to OD_600nm_ of 0.6; the RNA was isolated and reverse transcribed into cDNA. As indicated by dotted lines, log_2_ fold changes < −1 and > 1 were set as significant for differential gene expression. (**A**) Differential gene expression of *agaR*, *gadV, celA*, *eng*, and *spd_1588*. (**B**) Differential gene expression of the competence genes *comX1*, *comEA*, *comGA*, and *comD*. The log_2_ fold changes of differential gene expression in four replicates and their means are presented.

**Figure 4 microorganisms-09-00468-f004:**
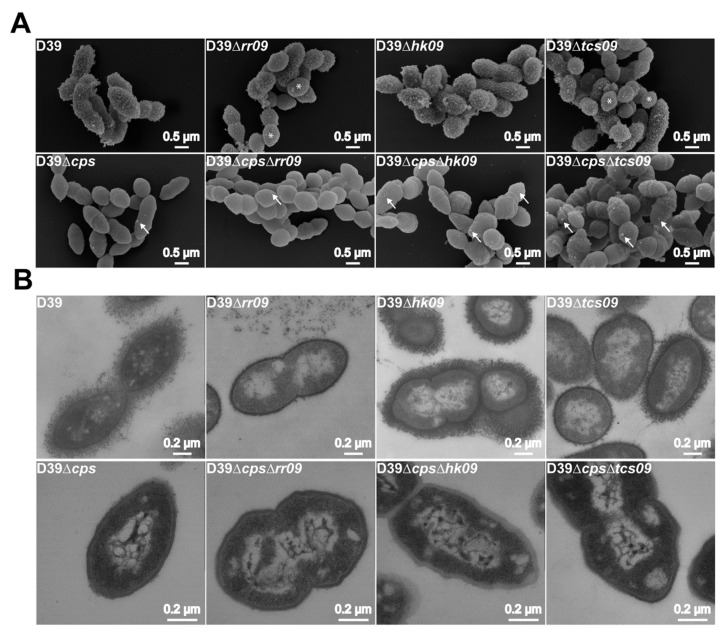
Influence of pneumococcal *tcs09*-mutations on pneumococcal cell morphology and capsule. (**A**) The pneumococcal cell morphology and capsular polysaccharide layer of D39, D39Δ*cps*, and their isogenic *rr09*-, *hk09*-, and *tcs09*-mutants illustrated by field emission scanning electron microscopy (FESEM). The white bars correspond to 500 nm. White arrows indicate outer membrane vesicles and white stars indicate cells with lower capsule amount. (**B**) TEM illustration of pneumococcal cell morphology and capsular polysaccharide layer. The white bars correspond to 200 nm.

**Figure 5 microorganisms-09-00468-f005:**
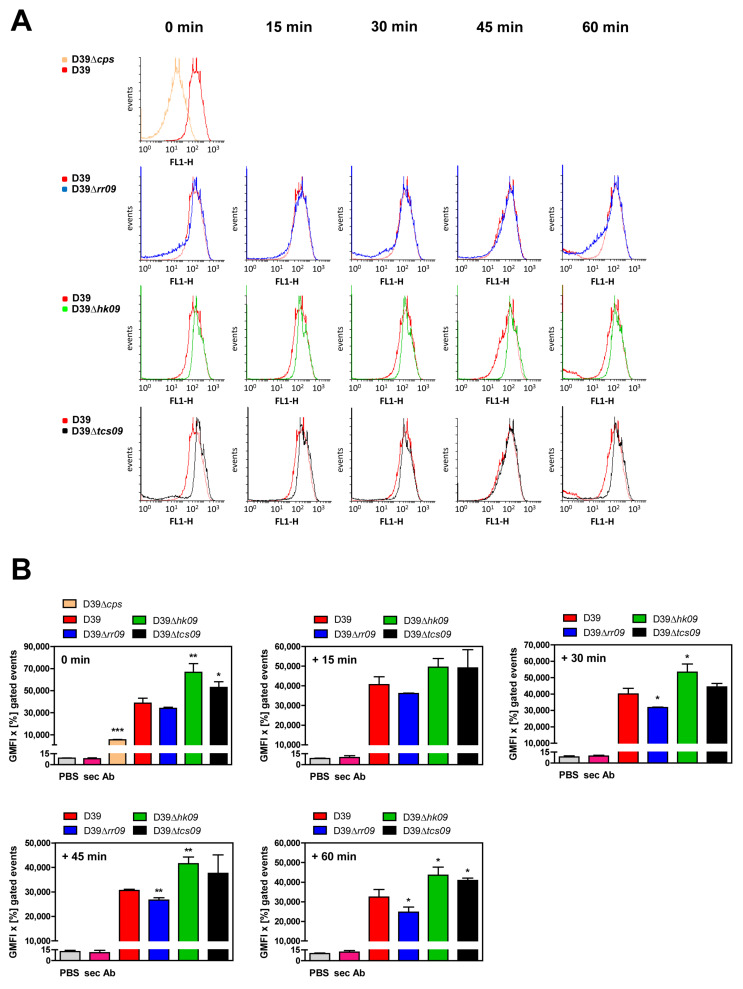
Abundance of surface exposed capsule in *S. pneumoniae* D39 and its isogenic *tcs09*-mutants measured via flow cytometry. The capsule expression of pneumococci was investigated via flow cytometry over a period of time of 60 min. For the analysis, 2 × 10^8^ bacteria of D39Δ*cps*, the D39 wild-type, and its isogenic Δ*rr09*-, Δ*hk09*-, and Δ*tcs09*-mutants cultured in CDM with glucose were used. To analyze capsule detachment, pneumococci were resuspended in PBS over 15–60 min before capsule detection. (**A**) An increase of the fluorescence intensity (FL-1 H channel) in the histograms shows a higher capsule expression of the pneumococci. (**B**) Shown are the calculated geometric mean fluorescence intensities (GMFI) x [%] gated events. Results are presented as the mean ± SD for three independent experiments. An independent student’s t-test proved a significance with *p*-values * < 0.05, ** < 0.01, and *** < 0.001 relative to the parental pneumococcal strain D39.

**Figure 6 microorganisms-09-00468-f006:**
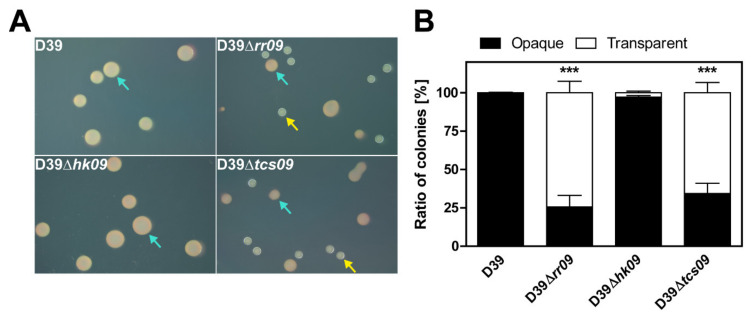
Colony morphology of opaque (O) and transparent (T) variants of *S. pneumoniae* D39 and isogenic *tcs09*-mutants. (**A**) Transparent (yellow arrow) and opaque (turquoise arrow) colonies of D39 and isogenic mutants plated on tryptic soy agar supplemented with catalase. (**B**) Percentage distribution of O and T colonies in D39 and isogenic mutants. Images were taken under oblique transmitted illumination using a Leica M125 C dissecting microscope and LAS X software. Results are presented as the mean ± SD for three independent experiments. An independent student’s *t*-test proved a significance with *p*-value *** < 0.001 relative to the parental pneumococcal strain D39.

**Figure 7 microorganisms-09-00468-f007:**
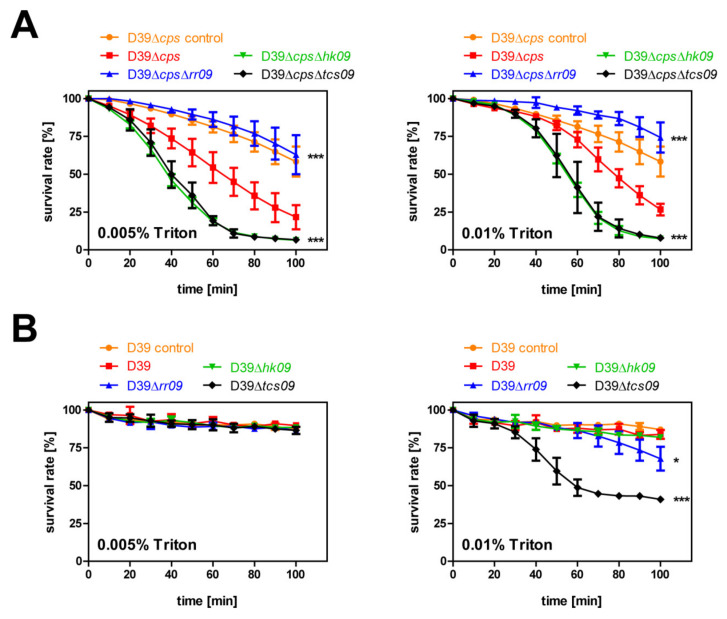
Effect of TCS09 in stability and autolysis in *S. pneumoniae*. The pneumococcal parental strains D39Δ*cps* or D39 and their isogenic mutants Δ*rr09*, Δ*hk09*, and Δ*tcs09* were cultivated in CDM until an OD_600nm_ 0.6, resuspended in PBS, and treated with Triton X-100. (**A**) Survival rate of nonencapsulated D39Δ*cps* and *tcs09*-mutant strains. (**B**) Percentage of survived encapsulated D39 and *tcs09*-mutant strains. Results are presented as the mean ± SD of the normalized percentage for three independent experiments. A two-way ANOVA proved a significance with a *p*-value * < 0.05 and *** < 0.001 relative to the parental pneumococcal strain.

**Figure 8 microorganisms-09-00468-f008:**
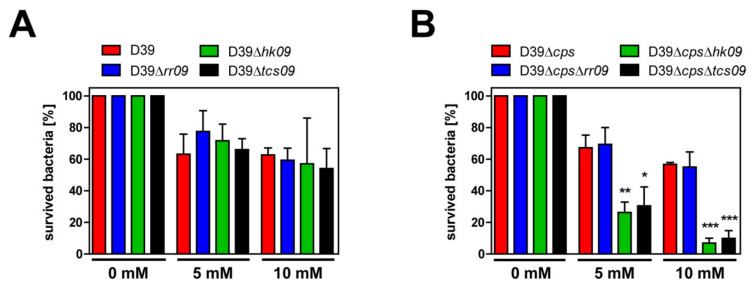
Resistance of pneumococcal *tcs09*-mutants against oxidative stress. Survivability of the *S. pneumoniae* strain D39, D39Δ*cps*, and its isogenic *rr09*-, *hk09*-, and *tcs09*-mutants under oxidative stress conditions. Pneumococci were cultured in CDM up to an OD_600nm_ of 0.6, and incubation with different H_2_O_2_ concentrations was performed for 30 min at 37 °C. The number of CFU were determined by plating a dilution series on blood agar. (**A**) Survival rate of encapsulated D39 and *tcs09*-mutant strains. (**B**) Percentage of survived bacteria of nonencapsulated D39Δ*cps* and *tcs09*-mutant strains. The measured values, presented as normalized percentage, are shown as mean ± SD of three independently performed experiments. An independent student’s *t*-test proved a significance with a *p*-value * < 0.05, ** < 0.01, and *** < 0.001 relative to the parental pneumococcal strain D39Δ*cps*.

**Table 1 microorganisms-09-00468-t001:** *S. pneumoniae* wild-type strains and mutants used in this study.

Strain	Capsule Type	Resistance	Knockout Genes	Reference
D39	2	-	-	[20]
D39Δ*rr09*	2	erythromycin	*spd_0574*	This study
D39Δ*hk09*	2	erythromycin	*spd_0575*	This study
D39Δ*tcs09*	2	erythromycin	*spd_0574, spd_0575*	This study
D39Δ*cps*	2	kanamycin	*spd_0312–spd_0333*	[21]
D39Δ*cps*Δ*rr09*	2	kanamycin, erythromycin	*spd_0312–spd_0333, spd_0574*	This study
D39Δ*cps*Δ*hk09*	2	kanamycin, erythromycin	*spd_0312–spd_0333, spd_0575*	This study
D39Δ*cps*Δ*tcs09*	2	kanamycin, erythromycin	*spd_0312–spd_0333 spd_0574, spd_0575*	This study

**Table 2 microorganisms-09-00468-t002:** Primers used for mutagenesis.

No.	Primer	Sequence 5′-3′	Restriction Site
100	MC-Erm-R	CCCGGGGAAATTTTGATATCGAT**AAGCTT**GAATTCCCGTAGGCGCTAGGGACCTC	*Hin*dIII
105	Erm Forw	GATGATGATGATCCCGGGTACC**AAGCTT**GAATTCACGGTTCGTGTTCGTGCTG	*Hin*dIII
106	Erm rev	AGTGAGTGAGTCCCGGGCTCGAG**AAGCTT**GAATTCGTAGGCGCTAGGGACCTC	*Hin*dIII
1122	*rr09*_Start Forw	CAGAACGAGCTTTCTCAAACC	
1123	*rr09_*End Rev	GCCTCCATTTTGCTAGACGA	
1124	*rr09_*End Forw	CTCACTG**AAGCTT**GAGATCGCAGAGAAGGTTGG	*Hin*dIII
1125	*rr09_*Start Rev	ATCATC**GCTAGC**GTAGGCTGCTACATTGACC	*Nhe*I
1126	*hk09_*Start Forw	GGTCAATGTAGCAGCCTACGA	
1127	*hk09_*End Rev	CGCACCTCCGATTAATTTTG	
1128	*hk09_*End Forw	CTCACTG**AAGCTT**CGATCAACGGCTCAAACTTC	*Hin*dIII
1129	*hk09_*Start Rev	ATCATC**GCTAGC**CAACCAGAGCTAGGAGAA	*Nhe*I
1232	*rr09_*sr_fw *Hin*dIII	ATCATC**AAGCTT**GTAGGCTGCTACATTGACC	*Hin*dIII
1234	*Nhe_erm* f	GCTT**GCTAGC**GACGGTTCGTGTTCGTGCTG	*Nhe*I

Restriction sites are shown in bold.

**Table 3 microorganisms-09-00468-t003:** Plasmids used for mutagenesis.

Plasmids	Properties	Source
pGXT	pGEM-T Easy backbone, linker region with toxic gene *ccdB* (Flanked by two *Xcm*I sites), TA cloning vector (3047 bp), Amp^R^, Pt7, *lacZ*	[22]
pGSP72N	pSP72 derivative, *Hin*dIII restriction site eliminated, generation of a *Nhe*I restriction site	This study
pSP72	Cloning vector (2462 bp), Amp^R^	Promega
pGXTΔ*rr09*::*erm*^R^	pGXT derivative vector with the subcloned 5′ and 3′-end homolog fragments of *spd_0574* interrupted by *ermB* resistance gene cassette for mutagenesis	This study
pGSP72NΔ*hk09*::*erm^R^*	pGSP72N derivative vector with the subcloned 5′ and 3′-end homolog fragments of *spd_0575* interrupted by *ermB* resistance gene cassette for mutagenesis	This study
pSP72Δ*tcs09*::*erm*^R^	pSP72 derivative vector with the subcloned 5′ end homolog fragment of *spd_0574* and 3′ end homolog fragment of *spd_0575* interrupted by *ermB* resistance gene cassette for mutagenesis	This study

**Table 4 microorganisms-09-00468-t004:** Primers used for qPCR.

Target Gene	Primer	Sequence 5′➔3′
*enolase* (*spd_1012*)	*eno*RT_F *eno*RT_R	CGGACGTGGTATGGTTCCA TAGCCAATGATAGCTTCAGCA
*agaR* (*spd_0064*)	qPCR *agaR* Forward qPCR *agaR* Reverse	TCATTTCAATGTACGATGTCAG GTTTGCGTGCACGTGAAACG
*gadV* (*spd_0066*)	qPCR *gadV* Forward qPCR *gadV* Reverse	ACGACGAAGTTGTCAACAACG GAGACGTTGGCTATCGTATTT
*celA* (*spd_0277*)	qPCR *celA* Forward qPCR *celA* Reverse	ATGTTATGACTGCTGGTCGTC CCATTTCAGCAAAAAGTGCTAT
*eng (spd_0335)*	qPCR *eng* Forward qPCR *eng* Reverse	GTCCGGTTCTTGCAGATAGC CCACCTTAGGCGCTTCAAAA
*spd_1588*	qPCR *spd_1588* Forward qPCR *spd_1588* Reverse	GTATTCATCTACTTAGCTGGAG ACATCACAACTAAAATGGATAATA

**Table 5 microorganisms-09-00468-t005:** Characteristic growth parameters of strains D39, D39∆*cps*, and isogenic mutants.

Strain	Growth Rate [h^−1^]	Max OD_600nm_	Generation Time ± SD [min]
**glucose**			
D39	0.5252	1.199	79.3 ± 4.3
D39∆*rr09*	0.5503	1.125	75.7 ± 2.6
D39∆*hk09*	0.3226	0.943	128.7 ± 8.4
D39∆*tcs09*	0.3859	1.173	107.3 ± 11.2
D39∆*cps*	0.3224	0.910	135.3 ± 36.8
D39∆*cps*∆*rr09*	0.3296	0.927	135.8 ± 46.3
D39∆*cps*∆*hk09*	0.3804	0.900	114.3 ± 35.6
D39∆*cps*∆*tcs09*	0.3664	0.880	114.5 ± 14.4
**galactose**			
D39	0.3174	1.436	131.2 ± 4.5
D39∆*rr09*	0.3546	1.313	117.4 ± 3.4
D39∆*hk09*	0.1878	0.909	277.4 ± 148.3
D39∆*tcs09*	0.2611	1.002	160.3 ± 14.4
D39∆*cps*	0.4614	1.073	90.1 ± 5.1
D39∆*cps*∆*rr09*	0.5917	1.027	70.2 ± 1.9
D39∆*cps*∆*hk09*	0.2830	0.251	148.6 ± 17.9
D39∆*cps*∆*tcs09*	0.2779	0.241	149.7 ± 5.2

## Data Availability

Data obtained from the RNA-seq analysis have been uploaded to the National Center for Biotechnology Information (NCBI) at the Gene Expression Omnibus (GEO) under accession number GSE165642 or this link: https://www.ncbi.nlm.nih.gov/geo/query/acc.cgi?acc=GSE165642, accessed on 1 December 2020.

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
