# Peer review of "The Two-Component System 09 Regulates Pneumococcal Carbohydrate Metabolism and Capsule Expression"

_microorganisms, 2021, doi:10.3390/microorganisms9030468_

Round 1

Reviewer 1 Report

Hirschmann et al. investigate the role of the two-component system 09 (TCS09) on pneumococcal pathophysiology using mutagenesis, microscopy, and transcriptomic analysis (RNA-Seq and qPCR). The study is novel, well designed, and executed. The study’s findings will enhance our understanding of the impact of the TCSs, specifically TCS09, on metabolic fitness, oxidative stress, and pneumococcal virulence. Below are some minor comments for the authors.

  1. The authors have specified the generic gene names involved in different pneumococcal TCSs. It would be great to include the names or locus tags of the genes involved in the TCS09 system in standard pneumococcal references, e.g., TIGR4 or D39.
  2. The authors should try to explain the differences in growth seen only in the encapsulated hk09-deficient and the wild-type encapsulated D39 strain in Glc media when no differences were observed between the encapsulated D39Dtcs09 and wild-type D39.
  3. It’s not clear what the sheet names, i.e., EIP-, IM-, CP-, GPI-, and UN-genes in the Table _S1 Excel supplementary data mean. These should define the terms in the table caption.
  4. The authors mention that the expression of some genes based on the RNA-Seq was validated using qPCR. How was the qPCR gene expression data normalized to account for sample heterogeneity, for example, different quality and quantity of the initial material?
  5. It would be great to include some volcano plots to show the differential gene expression patterns for different strains and growth conditions. It would also be great to provide data showing the expression patterns of all the genes in D39 (up/down-regulated and unchanged). The availability of such data would allow the reader to understand the gene expression patterns better.
  6. The rationale for studying the effect of the TCS09 on pneumococcal autolysis induced with Triton-X100 and oxidative stress is not clear to me.
  7. The authors should deposit the RNA-Seq data in a public nucleotide sequence database and include the associated accession numbers in the manuscript.

Author Response

Reviewer comments and authors point-by-point response:

Reviewer 1

Comment 1:

The authors have specified the generic gene names involved in different pneumococcal TCSs. It would be great to include the names or locus tags of the genes involved in the TCS09 system in standard pneumococcal references, e.g., TIGR4 or D39.

Response

We thank the reviewer for this suggestion and have included the names and identifier of the genes for S. pneumoniae strains D39 and TIGR4. D39: RR09: spd_0574,SPD_RS03105; HK09: spd_0575, SPD_RS03110. TIGR4: RR09: sp_0661, SP_RS03245; HK09: sp_0662, SP_RS03250.

The gene names and locus tags for D39 were included in the manuscript section Material & Methods (line 134).

Comment 2:

The authors should try to explain the differences in growth seen only in the encapsulated hk09-deficient and the wild-type encapsulated D39 strain in Glc media when no differences were observed between the encapsulated D39Dtcs09 and wild-type D39.

Response

We thank the reviewer for addressing this finding. The growth curve and generation time of D39Δtcs09 (g=107 min) in glucose supplemented medium was found to be between the growth curves and generation time of D39Δrr09 (g= 76 min) and D39Δhk09 (g=129 min). This indeed means that the loss of function of the complete TCS09 has only a minor influence on pneumococcal growth. While, our phenotypic analysis indicated a higher capsule amount of the D39Δtcs09 mutant compared to the wild-type, it also revealed that the planktonic culture of D39Δtcs09 is a mixture of low-encapsulated and high-encapsulated pneumococci. This is shown in Figure 5 and S6. Because the capsule production is highly energy dependent, we have a mix of faster growing low-encapsulated and slower growing high-encapsulated pneumococci. This probably results finally in the measured minor growth reduction. In contrast, analysis of the D39Δhk09 mutant revealed a homogenous population with significantly higher capsule amount compared to the wild-type D39 (Figures 5 and 6), which could explain the more pronounced growth defect of D39Δhk09.

We have addressed this point now in the Results (line 292-293) and Discussion (line 680-682).

Comment 3:

It’s not clear what the sheet names, i.e., EIP-, IM-, CP-, GPI-, and UN-genes in the Table _S1 Excel supplementary data mean. These should define the terms in the table caption.

Response

We apologize for not having included this important information. The abbreviations are now mentioned in Table S1 (line 1 of each sheet). EIP: Environmental Information Processing; IM: Intermediary Metabolism; CP: Cellular Processes; GIP: Genetic Information Processing; UN: Unknown Function

Comment 4:

The authors mention that the expression of some genes based on the RNA-Seq was validated using qPCR. How was the qPCR gene expression data normalized to account for sample heterogeneity, for example, different quality and quantity of the initial material?

Response

Data normalization in qPCR was performed by including an invariant endogenous control (reference gene). In this assay we used enolase (spd_1012) to correct for sample variation, regarding, for example, reverse transcriptase efficiency, because there were no significant changes in enolase expression as measured by the RNA-seq analysis (see table below; only for review). A defined amount of 20 ng/µl cDNA was used for each reaction (which is now mentioned in the Material and Method section, line 201-203 and line 205).

Table: Differential gene expression (RNA-seq) of enolase (spd_1012) in D39 rr09/hk09/tcs09-mutants.

strain

log2 fold change

p-value

D39∆rr09

0.0749

0.9066

D39∆hk09

0.5695

0.0016

D39∆tcs09

0.3279

0.1586

New Figure S2B.

RNA integrity check with the Agilent Bioanalyzer. Shown are the electrophoretic representations of the individual RNA samples used for qPCR. The curves consist of added marker and the individual ribosomal RNAs, which are used to calculate the RNA quality.

This Figure is now included in Figure S2 as panel B. The RNA integrity check indicated that the RNA of all strains has a high quality and that the generated qPCR data are valid.

Comment 5:

It would be great to include some volcano plots to show the differential gene expression patterns for different strains and growth conditions

Response

We have generated the suggested volcano plots and the new Figure is included in the Supplement as Figure S4.

Figure S4 Volcano plots of the differentially expressed genes of D39Δrr09, D39Δhk09 and D39Δtcs09 identified by RNA-seq. Histograms represent the two-dimensional distribution of identified genes by fold change and p-value. Genes with a –log10 padj-value ≥ 1.3 and a log2 fold change ≥ 1 or ≤ -1 were set significant. Significantly downregulated genes are shown in blue and significantly upregulated genes in red.

It would also be great to provide data showing the expression patterns of all the genes in D39 (up/down-regulated and unchanged). The availability of such data would allow the reader to understand the gene expression patterns better

Response:

These complete set of significantly regulated genes is listed in Table S1. The heat maps in Figure 2, Figure S3, new Figure S5, and the newly included volcano plot (Figure S4) show all the genes that are significantly differentially expressed. Adding a new and additional Table with the suggested information might lead to confusions in the view of the authors.

Comment 6:

The rationale for studying the effect of the TCS09 on pneumococcal autolysis induced with Triton-X100 is not clear to me.

Response:

Our FESEM studies suggested alterations in the cell morphology and cell wall integrity in D39ΔcpsΔhk09 and D39ΔcpsΔtcs09. Thus we hypothesized that these bacterial cells lyse faster under stress conditions. To accelerate or induce pneumococcal cell lysis, Triton X-100 can be used at a low concentration. Triton X-100 is a non-ionic detergent breaking protein-lipid and lipid-lipid associations, but does not influence protein-protein interaction or denature proteins. Polar head groups of Triton X-100 disrupt the hydrogen bonding in lipid bilayer, because it becomes inserted in the lipid bilayer and ultimately the integrity of the lipid membrane is disrupted. The integrity of the membrane is destroyed leading finally to (bacterial) cell lysis. Because of the cell wall alterations in the above mentioned mutants, lysis induced by Triton X-100 occures faster in the mutants.

Hydrogen peroxide, a ROS, diffuses, but limited, through the lipid bilayer of cell membrane (Seaver et al., 2001). This diffusion is dependent on changes in the membrane composition and channel proteins can promote diffusion (Bienert et al., 2006; Sousa-Lopes et al., 2004). Hydrogen peroxide can modulate the activity of different cellular components, for example: oxidation of DNA, proteins and membrane lipids resulting in accumulation of irreversible oxidative damages (Linley et al., 2012).

We see in our study at least two effects: i) hydrogen peroxide oxidizes membrane lipids which leads to irreversible membrane damage and lysis; ii) D39ΔcpsΔhk09 and D39ΔcpsΔtcs09 show cell wall alterations, which is probably associated with an altered cell wall integrity. This in turn might lead to a higher diffusion of hydrogen peroxide into the bacterium inducing the described accumulation of irreversible oxidative damages. Both effects result in cell lysis, which is promoted by the described phenotypes of Δhk09- and Δtcs09-mutants.

The reasons for studying the effect of the TCS09 on pneumococcal autolysis induced with Triton-X100 and facing the mutant to oxidative stress is now clearly mentioned in the result sections (line 499-505 and 533-537).

Seaver, L.C.; Imlay, J.A. Hydrogen Peroxide Fluxes and Compartmentalization inside Growing Escherichia coli. Journal of Bacteriology 2001, 183, 7182, doi:10.1128/JB.183.24.7182-7189.2001.

Bienert, G.P.; Schjoerring, J.K.; Jahn, T.P. Membrane transport of hydrogen peroxide. Biochimica et Biophysica Acta (BBA) - Biomembranes 2006, 1758, 994-1003, doi:https://doi.org/10.1016/j.bbamem.2006.02.015.

Sousa-Lopes, A.; Antunes, F.; Cyrne, L.; Marinho, H.S. Decreased cellular permeability to H2O2 protects Saccharomyces cerevisiae cells in stationary phase against oxidative stress. FEBS Letters 2004, 578, 152-156, doi:https://doi.org/10.1016/j.febslet.2004.10.090.

Linley, E.; Denyer, S.P.; McDonnell, G.; Simons, C.; Maillard, J.-Y. Use of hydrogen peroxide as a biocide: new consideration of its mechanisms of biocidal action. Journal of Antimicrobial Chemotherapy 2012, 67, 1589-1596, doi:10.1093/jac/dks129.

Comment 7:

The authors should deposit the RNA-Seq data in a public nucleotide sequence database and include the associated accession numbers in the manuscript

Response:

The data have been submitted and the data are available on GEO under the accession number GSE165642.

Reviewer 2 Report

This is a very comprehensive study examining the role of TCS09 in the physiology of Streptococcus pneumoniae strain D39. The authors have generated deletion mutants targeting individual elements of TCS09 (response regulator, rr09; his kinase, hk09) as well as entire TCS locus and have compared the transcriptome, growth, and phenotypic characteristics of this strain.

Overall the divergence of Δhk09 mutants from Δrr09 and Δtcs09 is a very interesting finding, which raises following questions that are not addressed by authors.

  1. At molecular levels, why is the Δhk09 mutant so different (in phenotype and gene expression) from Δtcs09 mutant which lacks both hk09 and rr09?
  2. Do tcs09 mutants (Δhk09, Δrr09, and Δtcs09) cause polar effects? This is very important especially given the close proximity of rr09 and hk09 in D39 genome. This can be resolved by complemented strains carrying a plasmid-borne copy of the deleted gene.
  3. Plasmid-borne rr09 (prr09) can be used to test whether Δtcs09/prr09 strain is similar to Δhk09
  4. Have the authors generated and analyzed double KO mutants (Δhk09Drr09) using KO primer pairs specific for Δhk09 and Δrr09?

Other major concerns include:

Is the generation time same as doubling time? The authors should provide average generation time ± standard deviation.

For figure 2B, showing individual replicate heat map will be more informative. This is important because in 2A, PCA shows that the replicates for Dtcs09 KO did not group together.

Minor comments:

For Table 2, please mention that the highlighted nucleotides represent R-endo sites.

Confirm that Fig 3 shows qRTPCR data.

Clarify that CPS stands for capsule.

What are the results of autolysis of Δcps strains in 0.01% TX100 and of capsulated strains in 0.005% TX100? Similarly, what are the results of oxidative stress response in capsulated strains (and KOs)?

Author Response

Reviewer comments and authors point-by-point response:

Reviewer 2

Comment 1:  At molecular levels, why is the Δhk09 mutant so different (in phenotype and gene expression) from Δtcs09 mutant which lacks both hk09 and rr09?

Response:

Because we do not know the exact role of the RR09 and HK09 and their interaction, we cannot predict accurately what the expected effects of each of the mutations will be. Many of the studies performed on bacterial TCS focus mostly on the RR and its targets, and do not analyze in detail the effects of all the mutants as we have done here. For instance, it is possible that the RR09 is modulating additional regulatory changes in the absence of the HK09, such as repression of transcription in its non-phosphorylated state. We also do not know the kinase/phosphatase kinetics of the HK09 and the in vivo phosphorylation state of the RR09. Interestingly, crosstalk among the different pneumococcal TCS was observed in other studies. This may represent a plausible explanation of the regulatory effect seen in our RNA-seq analysis. Furthermore, the signal for the HK09 is still unknown. Since the dimerization and histidine phosphotransfer domain is the one, which has been proven to be involved in the specificity between HK and RR, it would be of great interest to analyze how specific the HK-RR09 couple is. Unfortunately, such studies have not been performed for S. pneumoniae and because every bacterium is unique and independent on its regulatory systems, this needs additional efforts.

We have addressed this point now in the Discussion (line 742-747).

Comment 2:

Do tcs09 mutants (Δhk09, Δrr09, and Δtcs09) cause polar effects? This is very important especially given the close proximity of rr09 and hk09 in D39 genome. This can be resolved by complemented strains carrying a plasmid-borne copy of the deleted gene.

Response:

The reviewer mentions an important aspect that we had checked after generating the mutants. We did not observe any polar effects in our generated mutants. The genes downstream (or upstream) of the genes knocked-out by the insertion-depletion strategy are continuously expressed and their expression level is not changed. This is clearly shown in the RNA-seq data (log2 fold change RNA expression pattern) and in the figure of the RNA-seq profiles (only for review) below. This is now mentioned in line 161-164 in the manuscript.

Table: Differential gene expression (RNA-seq) of rr09 (spd_0574) and hk09 (spd_0575) in D39 tcs09-mutants.

strain

log2 fold change

p-value

rr09

D39∆rr09

-8.5732

5.8765E-80

D39∆hk09

0.0115

0.9387

D39∆tcs09

-7.7892

2.1978E-149

hk09

D39∆rr09

0.5482

0.0168

D39∆hk09

-7.3228

1.9948E-202

D39∆tcs09

-8.0756

3.2655E-230

Figure (only for review9: RNA expression pattern of rr09, hk09, spd_0576 and zmpB in D39 and corresponding tcs09-mutants obtaines by RNA-seq.

Comment 3:

Plasmid-borne rr09 (prr09) can be used to test whether Δtcs09/prr09 strain is similar to Δhk09

Response:

This is a very nice suggestion, but regarding the genetics of pneumococci not so easy to conduct due to the following points: 1) this would require only one copy of the plasmid per bacterial cell; 2) the native promotor is needed to have a similar strength and the so far used promotors are strong promotors thereby changing the gene expression; 3) if we change the copy number of the regulator in a bacterial cell this will probably lead to competitions for other binding sites; 4) these effect might result in an artificial physiology; 5) our own experience with in trans complementations shows that not all complementations work properly, the reasons are still unknown

Comment 4:

Have the authors generated and analyzed double KO mutants (Δhk09Drr09) using KO primer pairs specific for Δhk09 and Δrr09

Response

In a routine process we always check our mutant on the molecular levels at least by PCR. If possible, mutants are also checked at the protein level using specific antibodies, which was not possible here. The mutants were confirmed on the DNA level by specific PCR reactions using different primer combination and on the RNA-level by RNA-seq. (see line 138 – 139 in the manuscript)

Figure S1: Genomic organization of the tcs09 gene cluster in S. pneumoniae D39 wild-type and isogenic tcs09-mutants. Here the rr09 gene is shown in blue and the hk09 gene in green within its operon organization, the genes upstream and downstream of rr09 and hk09 are shown in grey and the inserted ermB- cassette for tcs09 deletions is indicated by orange. The large and filled arrows represent their relative gene size and orientation in the genome. Transcriptional start site is indicated with a thin arrow and terminators with lollypops.

The mutagenesis scheme is referred to in line 135 in the manuscript and added as new Figure S1.

Other comments of Reviewer 2:

  1. Is the generation time same as doubling time? The authors should provide average generation time ± standard deviation.

Response

Indeed, the generation time is the time it takes for the population to double, also called doubling time. Another column with the generation time ±SD [min] is now inserted in Table 5.

Table 5. Characteristic growth parameters of strains D39, D39∆cps and isogenic mutants.

strain

growth rate [h-1]

max OD600nm

generation time ± SD [min]

glucose

D39

0.5252

1.199

79.3 ± 4.3

D39∆rr09

0.5503

1.125

75.7 ± 2.6

D39∆hk09

0.3226

0.943

128.7 ± 8.4

D39∆tcs09

0.3859

1.173

107.3 ± 11.2

D39∆cps

0.3224

0.910

135.3 ± 36.8

D39∆cps∆rr09

0.3296

0.927

135.8 ± 46.3

D39∆cps∆hk09

0.3804

0.900

114.3 ± 35.6

D39∆cps∆tcs09

0.3664

0.880

114.5 ± 14.4

galactose

D39

0.3174

1.436

131.2 ± 4.5

D39∆rr09

0.3546

1.313

117.4 ± 3.4

D39∆hk09

0.1878

0.909

277.4 ± 148.3

D39∆tcs09

0.2611

1.002

160.3 ± 14.4

D39∆cps

0.4614

1.073

90.1 ± 5.1

D39∆cps∆rr09

0.5917

1.027

70.2 ± 1.9

D39∆cps∆hk09

0.2830

0.251

148.6 ± 17.9

D39∆cps∆tcs09

0.2779

0.241

149.7 ± 5.2

  1. For figure 2B, showing individual replicate heat map will be more informative. This is important because in 2A, PCA shows that the replicates for Dtcs09KO did not group together.

Response:

We have created a heatmap showing the individual replicates as suggested by the reviewer. These data are now shown in the supplement in Figure S5.

3 .For Table 2, please mention that the highlighted nucleotides represent R-endo sites

Response:

The explanation for nucleotides shown in bold is added below Table 2

  1. Confirm that Fig 3 shows qRTPCR data.

Response:

We apologize that this was not clear. The figure description was modified and now reads: “Figure 3. Differential gene expression in S. pneumoniae D39 and isogenic tcs09-mutants using qPCR.” (lines 392-393).

  1. Clarify that CPS stands for capsule.

Response:

This information is now included in the Material and Method section; nonencapsulated means Δcps. Furthermore it was clarified, that CPS stands for “capsular polysaccharide”. (line 245 in Material and Method section).

  1. What are the results of autolysis of Δcps strains in 0.01% and capsulated strains in 0.005% TX100?

Response:

These data were generated and included in the revised version. We observed a dose-dependent effect in encapsulated strains with a faster lysis of D39Δrr09 and D39Δtcs09 when using higher concentrations of Triton X-100 and no significant dose-dependent effect in nonencapsulated wild-type and tcs09-mutant strains. The figure and results description were adjusted in the revised version as follows: “Our FESEM studies suggested alterations in the cell morphology and cell wall integrity in D39ΔcpsΔhk09 and D39ΔcpsΔtcs09 (Figure 4A). Thus we hypothesized that these bacterial cells lyse faster under stress conditions. To accelerate or induce pneumococcal cell lysis, Triton X-100 was used at a low concentration to study the impact of the TCS09 on pneumococcal stability and autolysis. Triton X-100 is a non-ionic detergent with polar head group disrupting the hydrogen bonds in lipid bilayers. The integrity of the lipid membrane is disrupted, which ultimately leads to cell lysis.” (lines 499-505).

Figure 7. Effect of TCS09 in stability and autolysis in S. pneumoniae. The pneumococcal parental strains D39Δcps or D39 and their isogenic mutants Δrr09, Δhk09 and Δtcs09 were cultivated in CDM until an OD600nm 0.6, resuspended in PBS and treated with Triton X-100. (A) Survival rate of nonencapsulated D39Δcps and tcs09–mutant strains. (B) Percentage of survived encapsulated D39 and tcs09-mutant strains. Results are presented as the mean ± SD of the normalized percentage for three independent experiments. A two-way Anova proved a significance with a p-value * < 0.05 and *** < 0.001 relative to the parental pneumococcal strain.

  1. What are the results of oxidative stress response in capsulated strains (and KOs)?

Response

These data were generated and included in the revised version. We detected no significant difference between encapsulated wild-type D39 and tcs09-mutants. The observed different survival rate in the nonencapsulated tcs09-mutants is probably a direct result of cell wall alteration as shown by FESEM. The figure and results description were adjusted in the revised version as follows: “Reactive oxygen species (ROS) like hydrogen peroxide can diffuse in a limited manner through the lipid bilayer of a cell membrane and cause oxidation of DNA, proteins, and membrane lipids resulting in accumulation of irreversible oxidative damages and cell lysis [1-4]. Because hk09- and tcs09-mutants showed alterations of the cell wall (Figure 4A), hydrogen peroxide might diffuse easier into the cell and cause cell damage and lysis.” (lines 533-537)

Figure 8. Resistance of pneumococcal tcs09-mutants against oxidative stress. Survivability of the S. pneumoniae strain D39, D39Δcps and its isogenic rr09-, hk09- and tcs09- mutants under oxidative stress conditions. Pneumococci were cultured in CDM up to an OD600nm of 0.6 and incubation with different H2O2 concentrations was performed for 30 min at 37°C. The number of CFU were determined by plating dilution series on blood agar. (A) Survival rate of encapsulated D39 and tcs09-mutant strains. (B) Percentage of survived bacteria of nonencapsulated D39Δcps and tcs09-mutant strains. The measured values, presented as normalized percentage, are shown as mean ± SD of three independently performed experiments. An independent student’s t-test proved a significance with a p-value * < 0.05, ** < 0.01 and *** < 0.001 relative to the parental pneumococcal strain D39Δcps.

  1. Bienert, G.P.; Schjoerring, J.K.; Jahn, T.P. Membrane transport of hydrogen peroxide. Biochimica et Biophysica Acta (BBA) - Biomembranes 2006, 1758, 994-1003, doi:https://doi.org/10.1016/j.bbamem.2006.02.015.
  2. Linley, E.; Denyer, S.P.; McDonnell, G.; Simons, C.; Maillard, J.-Y. Use of hydrogen peroxide as a biocide: new consideration of its mechanisms of biocidal action. Journal of Antimicrobial Chemotherapy 2012, 67, 1589-1596, doi:10.1093/jac/dks129.
  3. Seaver, L.C.; Imlay, J.A. Hydrogen Peroxide Fluxes and Compartmentalization inside Growing Escherichia coli. Journal of Bacteriology 2001, 183, 7182, doi:10.1128/JB.183.24.7182-7189.2001.
  4. Sousa-Lopes, A.; Antunes, F.; Cyrne, L.; Marinho, H.S. Decreased cellular permeability to H2O2 protects Saccharomyces cerevisiae cells in stationary phase against oxidative stress. FEBS Letters 2004, 578, 152-156, doi:https://doi.org/10.1016/j.febslet.2004.10.090.

Round 2

Reviewer 2 Report

First, I thank the authors for meticulously (and patiently!) addressing my concerns either with additional data or explanation in discussion section. A minor suggestion for the edited version is below:

In the discussion section, please explain why two of D39∆tcs09 replicates showed different expression within the competence cluster (shown in Fig S3). If you do not have concrete experimental evidence to support their explanation, an educated speculation would suffice!  

Author Response

Response

We appreciate the suggestion of the reviewer and have added the following paragraph to the discussion:

“Two replicates of the D39Dtcs09 mutant showed increased expression of competence genes in the RNA-seq analysis, most probably indicating higher extracellular levels of the competence-stimulating peptide (CSP) and concomitant activation of the ComE and ComX regulons. Because of the almost identical growth of the four replicates, this observation was hard to explain, but did not seem to be directly linked to TCS09 deficiency. Accordingly, qPCR-based analysis of independently generated samples confirmed that competence gene expression in D39Dtcs09cultures is consistent and comparable to the D39 wild-type”. (line 633-639).
